# In-plane hyperbolic polariton tuners in terahertz and long-wave infrared regimes

Wuchao Huang[1,6], Thomas G. Folland[2,3,6], Fengsheng Sun[1,6], Zebo Zheng [1,6], Ningsheng Xu[1,4], Qiaoxia Xing[5], Jingyao Jiang[1], Huanjun Chen [1] ✉, Joshua D. Caldwell [2] ✉, Hugen Yan [5] ✉ & Shaozhi Deng[1] ✉

One of the main bottlenecks in the development of terahertz (THz) and long-wave infrared (LWIR) technologies is the limited intrinsic response of traditional materials. Hyperbolic phonon polaritons (HPhPs) of van der Waals semiconductors couple strongly with THz and LWIR radiation. However, the mismatch of photon – polariton momentum makes far-field excitation of HPhPs challenging. Here, we propose an In-Plane Hyperbolic Polariton Tuner that is based on patterning van der Waals semiconductors, here α-MoO₃, into ribbon arrays. We demonstrate that such tuners respond directly to far-field excitation and give rise to LWIR and THz resonances with high quality factors up to 300, which are strongly dependent on in-plane hyperbolic polariton of the patterned α-MoO₃. We further show that with this tuner, intensity regulation of reflected and transmitted electromagnetic waves, as well as their wavelength and polarization selection can be achieved. Our results can help the development of THz and LWIR miniaturized devices.

The discovery of two-dimensional (2D) vdW crystals has opened avenues for exploring functional materials and devices in the THz (30–3000 μm) and long-wave infrared (LWIR; 8–15 μm) spectral ranges[1–10]. The THz and LWIR technologies are of great significance for future photonic and optoelectronic applications, such as 5 G/6 G mobile net-works[11,12], night vision[13], biomedical imaging and sensing[14,15], thermal management[16], and deep-space exploration[17]. However, their development is always limited by the scarcity of materials with strong and tunable intrinsic optical responses, in particular those used for devices in nanoscale and of room temperature operation.

In the past decades, much effort has been devoted to develop narrow band-gap semiconductors (e.g., mercury cadmium telluride and InSb) and quantum materials (e.g., quantum wells/dots, super lattice)[18–20], whose inter-band, intra-band, or inter-subband optical transitions are found in the LWIR and THz regimes. However, due to

the relatively weak light-matter operation, their optical responses are weak, and this is further affected by thermal noise. Thus their devices usually require a cryogenic operation to suppress thermal noise, and the introduction of components such as antennas and/or light absorbing layers to improve the electromagnetic absorption, the dimension of which will be out of the scale of tens of micrometers, not to mention the nanoscale. These will result in complex and large-volume architectures that are not favorable for nanodevices, even miniaturized and portable devices. Moreover, the broadband- and polarization-insensitive optical transitions of these materials also restrict their applications in spectrally and polarization-selective photonic and optoelectronic devices, which are in particular interesting in a modern information society. Metamaterials and metasurfaces comprised of artificially designed metallic or dielectric unit cells are able to confine the THz and LWIR electromagnetic waves to enhance light–matter interactions, which therefore give rise to a variety of

[1]State Key Laboratory of Optoelectronic Materials and Technologies, Guangdong Province Key Laboratory of Display Material and Technology, School of Electronics and Information Technology, Sun Yat-sen University, Guangzhou 510275, China. [2]Department of Mechanical Engineering, Vanderbilt University, Nashville, TN 37235, USA. [3]Department of Physics and Astronomy, The University of Iowa, Iowa City, IA 52245, USA. [4]The Frontier Institute of Chip and System, Fudan University, Shanghai 200433, China. [5]State Key Laboratory of Surface Physics, Department of Physics, Key Laboratory of Micro and Nano-Photonic Structures (Ministry of Education), Fudan University, Shanghai 200433, China. [6]These authors contributed equally: Wuchao Huang, Thomas G. Folland, Fengsheng Sun, Zebo Zheng. ✉e-mail: chenhj8@mail.sysu.edu.cn; Josh.caldwell@vanderbilt.edu; hgyan@fudan.edu.cn; stsdsz@mail.sysu.edu.cn

functional devices[21–26]. However, despite intense research efforts, in these spectral ranges, the unit cells of many conventional metamaterial/metasurface often offer restricted confinement due to high losses[21,27]. These are not conducive to device integration, and will also increase device power consumption.

Recently, semiconducting vdW transition metal oxides, such as α-MoO$_3$[8–10] and α-V$_2$O$_5$[28], have been explored to exhibit phonon polaritons−quasiparticles formed by the coupling of photons to phonons −with ultra-low-loss in the THz and LWIR regimes[6,28–35]. Due to the biaxial nature, in each Reststrahlen band bracketed by the longitudinal optical (LO) and transverse optical (TO) phonon frequencies, the real parts of the permittivities, Re($\varepsilon$), along the three optical principal axes in these crystals are different, and there is always at least one negative component[8–10,28,30,36]. This means that the α-MoO$_3$ and α-V$_2$O$_5$ can sustain natural in-plane HPhPs, enabling ultrahigh confinement and manipulation of the THz and LWIR radiation at nanoscale dimensions[31–39].

However, so far, the demonstration of utilizing the above exotic characteristics in a practical device is not given. The challenge lies in that one needs to compensate for the large photon−polariton momentum mismatch for far-field excitation and far-field characterization of the HPhPs. In the previous studies, the HPhPs of vdW crystals have been observed by near-field nano-imaging techniques[8–10,28], relying on using a metallic nanotip to compensate for the large momentum mismatch between free-space photons and polaritons. For most practical device applications, direct excitation of the HPhPs from the far-field is necessary. Some earlier studies indicate that it is possible to pattern the surface of vdW crystals, such as with graphene[40–42], hexagonal boron nitride[7,43,44], semi-metals[45], and topological insulators[46] to excite and measure the various types of polaritons. However, these previous studies focused on plasmon polaritons with in-plane isotropic[40–42,46] and hyperbolic dispersions[45], as well as phonon polaritons with in-plane isotropic dispersion[7,43,44]. Far-field excitation and characterization of the tunable in-plane HPhPs in vdW crystals, especially in the THz spectral regime, remain unexplored. Furthermore, exploring the applications of the in-plane HPhPs in optical devices has so far remained elusive. These can be done if one has a vdW crystal with a large enough lateral size while maintaining thicknesses of nanometer scale, so that patterns are made larger than the diffraction limit for these free-space wavelengths and, thus, suitable for far-field spectroscopy.

In this article, we demonstrate far-field excitation and far-field characterization of HPhPs in an In-Plane Hyperbolic Polariton Tuner, which is formed by patterning one-dimensional (1D) ribbon array directly onto the semiconducting HPhP vdW α-MoO$_3$ flake with a centimeter lateral size while maintaining thicknesses of 100-200 nm. The THz and LWIR photons from far-field illuminating onto the tuner will strongly couple with the phonons and give rise to polaritons with in-plane hyperbolicities. This makes an in-plane tuner an actual device that acts not only with functions of grating but also as a polarizer and notch filter in the LWIR and THz regimes, and have distinctive features including high-Q (300) resonance and extinction ratios up to 6.5 dB at a deep sub-wavelength thickness of 200 nm. Moreover, the polariton resonance frequency, i.e., the operation frequency of the polarizers and notch filters, can be highly tuned by varying the period and the skew angle of the ribbon array.

## Results

### Fabrication of tuners and far-field HPhPs excitation

An in-plane tuner will have a structured surface written with desirable patterns. In this study, the in-plane tuner consists of simple one-dimensional periodic ribbon patterns (1D-PRPs) directly formed on a vdW α-MoO$_3$ flake using electron-beam lithography (EBL). They have widths ($w$) and skew angles ($\theta$), which is defined by the angle between the long-axis ribbon direction and [001] crystallographic axis of vdW α-

MoO$_3$ (Fig. 1a). The ribbon period ($\Lambda$) is set as $2w$. Both of $w$ and $\Lambda$ are much smaller than the excitation wavelength. As such, we synthesized a 120 nm-thick α-MoO$_3$ crystal with a largest lateral size larger than 1 cm (Fig. 1b and Supplementary Fig. 1, and see Methods for details), which guarantees the fabrication, characterization, and comparison of different tuners on the same flake (Fig. 1c and Supplementary Fig. 2). The basal plane of the as-grown α-MoO$_3$ is (010) plane, with the two orthogonal directions corresponding to [100] and [001] crystallographic axes, respectively[8–10]. In our study, these two axes are defined as the x- and y-axes, respectively (Fig. 1a), which are identified experimentally using micro-Raman spectroscopy (Supplementary Fig. 1b–d). A broadband THz and LWIR light illuminates the tuner and the optical responses at the far-field were measured using a polarized Fourier transform infrared (FTIR) micro-spectroscopy (Fig. 1a) (see Methods for details). It should be noted that usually, three main techniques are employed for determining the broadband polaritonic properties of 2D crystals, including the FTIR[36,47,48], electron energy loss spectroscopy (EELS)[49], and infrared nanoscopy[4,9,28]. In comparison with the latter two techniques, which usually require complex instrumentation, harsh sample preparation, and time consumption, the far-field polariton characteristics of the 1D-PRPs can be readily measured in a common broadband FTIR spectrometer at ambient conditions, with low time-consuming, a high collection efficiency, and over a large sample area.

In vdW α-MoO$_3$ crystal, in the spectral regimes 230−400 cm$^{-1}$ (THz) and 545−1010 cm$^{-1}$ (LWIR), there are a series of Reststrahlen bands where the Re($\varepsilon$) along one of the three crystallographic axes, i.e., [100], [001], and [010], is negative (Supplementary Fig. 3), while at least one is positive. This makes α-MoO$_3$ a natural hyperbolic medium capable of supporting HPhPs. We first characterized the far-field reflection of the homogeneous pristine α-MoO$_3$, which is calculated as $R/R_0 - 1$, with $R$ and $R_0$ the reflectance of the light from the surfaces of the sample and bare substrate (see Methods). For incident light polarized along the [001] and [100] directions, the reflectance spectra show distinct peaks at 550 and 820 cm$^{-1}$ (Fig. 1d), which correspond to IR-active TO phonon modes along [001] ($\omega_{TO}^{001}$) and [100] axes ($\omega_{TO}^{100}$), respectively. Both of the spectra exhibit small valleys at 1004 cm$^{-1}$. These valleys are very close to the frequency of LO phonon mode along the [010] axis (z-axis, $\omega_{LO}^{010}$) where the permittivity diminishes. For a 120 nm-thick α-MoO$_3$ flake, leaky modes (Berreman modes) can be excited near this epsilon-near-zero (ENZ) region and then give rise to the two valleys on the reflectance spectra[50,51]. These narrow leaky modes can further interfere with the broad reflection background and generate asymmetric Fano lineshapes (see Supplementary Note 1 and Supplementary Fig. 4). However, no evident spectral peaks corresponding to HPhPs are observed in the rest of the spectral range. This is due to the large wavevector mismatch between free-space photons ($k_0$) and polaritons ($q_{PhPs}$), which prevents the coupling of electromagnetic fields to HPhPs.

A tuner comprised of 1D-PRPs (Fig. 1c and Supplementary Fig. 2) is able to overcome the large momentum mismatch and excite the HPhPs[52,53]. The incident waves will be scattered by the sharp ribbon edges into evanescent waves with large momenta, whereby HPhPs propagating transverse to the ribbons are excited. Fabry−Pérot resonances (FPRs) can then be formed upon the multiple polariton reflections from the ribbon edges. Simultaneously, the 1D-PRPs can also diffract the incident light into guided waves propagating perpendicular to the ribbon long axis, whose wavevectors are much larger than the free-space waves[54]. These guided waves can then couple with and transfer energy to the polariton FPRs (see Supplementary Note 2 and Supplementary Figs. 5, 6 for more discussion on the excitation of HPhPs by the 1D-PRP). The polariton energy will be dissipated by the lattice vibrations or radiated back to the free space, as manifested by resonance peaks and valleys in the corresponding reflectance and transmission spectra, respectively.

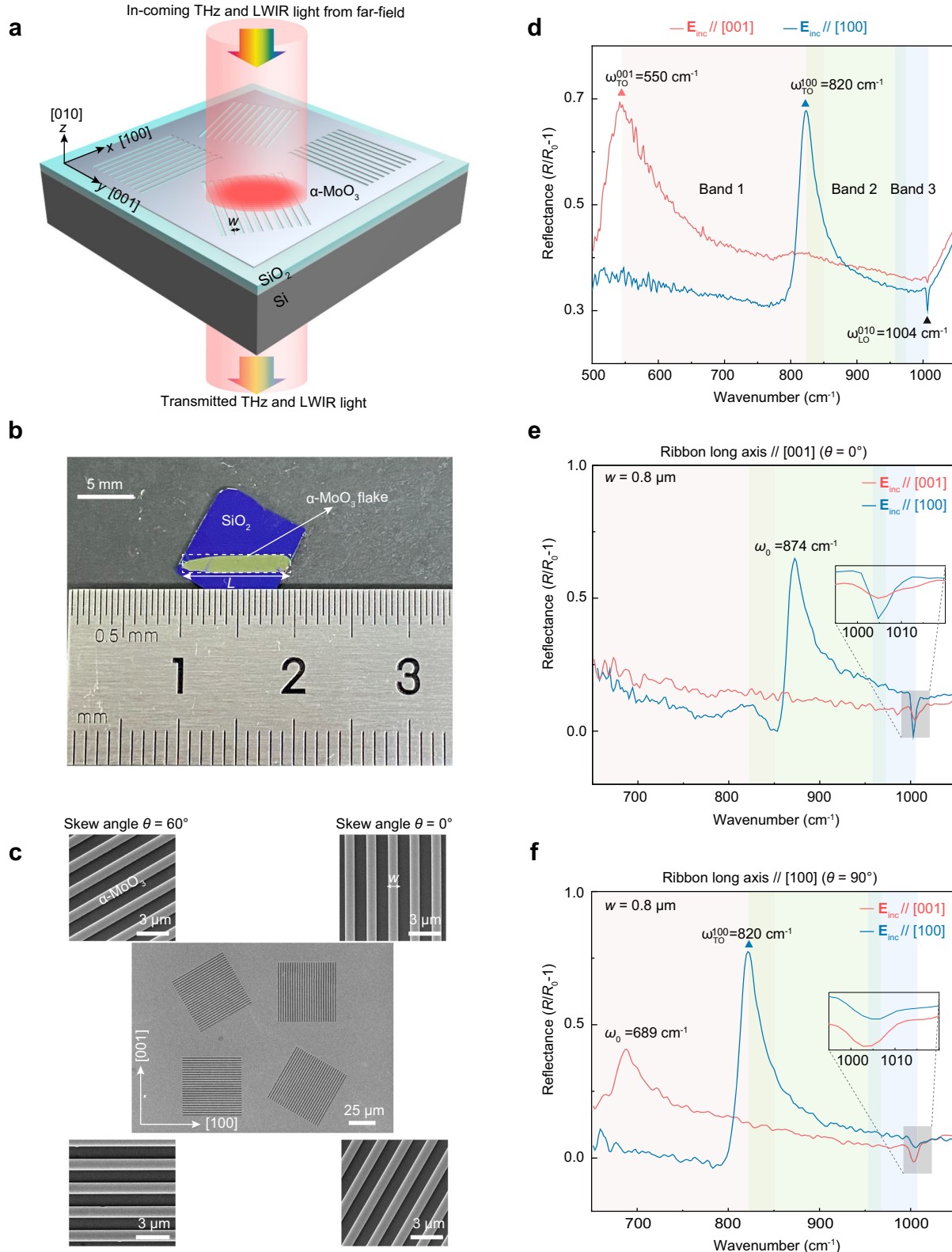

For a typical 1D-PRP with $w = 800$ nm and orientated along the [001] axis (sample with skew angle $\theta = 0°$ shown in Fig. 1c), HPhPs in Reststrahlen Band 2 (820 to 972 cm$^{-1}$, where Re($\varepsilon_x$) <0 and Re($\varepsilon_y$), Re($\varepsilon_z$) >0) will be excited upon illumination polarized along the [100] axis. This will lead to a strong reflectance peak at 874 cm$^{-1}$, as shown in Fig. 1e. The small bump at 820 cm$^{-1}$ originates from the intrinsic $\omega_{TO}^{100}$. No resonance peaks are observed when the polarization is switched to

[001] direction. In contrast, for 1D-PRP orientated along the [100] axis (sample with $\theta = 90°$ shown in Fig. 1c), a clear peak at 689 cm$^{-1}$ is identified (Fig. 1f), suggesting the launching of HPhPs in Reststrahlen Band 1 (545 to 851 cm$^{-1}$, where Re($\varepsilon_y$) <0 and Re($\varepsilon_x$), Re($\varepsilon_z$) >0). For polarization along the [100] axis, only the $\omega_{TO}^{100}$ peak appears. The excitation of HPhPs in both Bands 1 and 2 using these two types of 1D-PRPs can be further confirmed by simulating the near-field

**Fig. 1 | Far-field excitations of hyperbolic phonon polaritons (HPhPs) in in-plane hyperbolic polariton tuners. a** Schemes of the tuners comprised of van der Waals (vdW) α-MoO₃ periodic ribbon patterns and Fourier transform infrared spectroscopic (FTIR) measurements. **b** Photograph of a typical α-MoO₃ flake grown on a silicon substrate with a 300-nm oxide layer. The largest lateral length of the flake is 1 cm. **c** Scanning electron microscopy image of periodic ribbon patterns with a fixed width $w = 800$ nm and different skew angles $\theta$ of 0°, 30°, 60°, and 90°. **d** Polarized reflectance spectra of the pristine α-MoO₃ thin flake shown in (**b**). The electric fields of the incident light, $\mathbf{E}_{inc}$, are along [001] (red curve) and [100] (blue curve) crystallographic directions, respectively. **e, f** Experimental polarized reflectance spectra of one-dimensional periodic tuner patterns with $\theta = 0°$ (**e**) and 90° (**f**). The polarization of the incident light is paralleled to [001] (red curves) and [100] (blue curves) directions, respectively. Insets: enlarged reflectance spectra in the range of 995 to 1020 cm⁻¹. The $\omega_0$ represents the resonant frequency of polariton tuners. The $\omega_{TO}^{001}$ and $\omega_{TO}^{100}$ represent the transverse optical (TO) phonon frequencies along the [001] and [100] crystallographic directions, respectively, and the $\omega_{LO}^{010}$ indicates the longitudinal optical (LO) phonon frequency along the [010] crystallographic direction. Colored shaded regions in (**d**), (**e**), and (**f**) mark the frequency ranges of different Reststrahlen bands of α-MoO₃.

distributions they support. The simulated field distributions at $\omega = 874$ and 689 cm⁻¹ clearly reveal that polaritonic rays with zig−zag shape patterns propagate inside the ribbons (Supplementary Fig. 7a, b and Supplementary Note 3). These are typical fingerprints of HPhP wave-guide modes. These modes are bulk modes with electromagnetic fields confined inside the body of the flakes, such as those observed in hBN nanostructures[55] and biaxial α-MoO₃ flakes[56].

Notably, in the two 1D-PRPs two valleys with narrow linewidths appear around 1000 cm⁻¹ for both polarization conditions. These valleys are spectrally close, but occur with different amplitudes, with the spectral valleys deeper when the incident light is polarized perpendicular to the ribbon. In addition, as discussed below, the deeper valleys shift when $w$ and $\theta$ change. Therefore, they are ascribed to HPhP resonances in Reststrahlen Band 3 (958 to 1010 cm⁻¹, where $\text{Re}(\varepsilon_z) < 0$ and $\text{Re}(\varepsilon_x)$, $\text{Re}(\varepsilon_y) > 0$). This is also corroborated by the corresponding near-field distributions showing polaritonic rays with zig−zag shapes (Supplementary Fig. 7c, d). Due to their narrow linewidths, Fano interference will occur between the background reflectance and the HPhP resonances, giving rise to these valley features (see Supplementary Note 1 and Supplementary Fig. 4). The shallower valleys are contributed by the aforementioned ENZ condition that occurs near the $\omega_{LO}^{010}$ (Fig. 1d). These results are consistent with polariton propagation in the basal plane of an α-MoO₃ flake: the HPhPs in Band 1 and 2 are of in-plane hyperbolicity, which cannot propagate along [100] and [001] directions, respectively[10]. In contrast, the dispersion of HPhPs in Band 3 are elliptical, which therefore allows them to propagate along both of the two orthogonal crystallographic directions[10].

## Tuning the HPhPs with tuners of different ribbon widths

The above results clearly prove that the HPhPs supported by α-MoO₃, which previously were only accessed using near-field nano-imaging[8–10,30,31], are able to be excited from the far-field using the 1D-PRPs. To demonstrate the tunability of HPhPs in this material, we measured the reflectance spectra of 1D-PRPs with different $w$ ranging from 100 to 2000 nm. All spectra were collected with the incident light polarized perpendicular to the ribbon long axis. For 1D-PRPs parallel to the [100] direction, both of the HPhP resonances corresponding to Band 1 (peaks) and 3 (valleys) can be excited (Fig. 2a). Notably, when $w$ increases from 600 to 2000 nm, the resonance in Band 1 clearly redshifts from 746 to 613 cm⁻¹, whereas a reverse trend appears for the modes in Band 3, where the resonances blueshift from 998 to 1005 cm⁻¹. Similar spectral evolution with changing $w$ can be observed for 1D-PRPs along the [001] direction (Fig. 2b), where PhP resonances in Band 2 and 3 are excited. Additionally, the variation of resonance frequency in Band 3 with $w$ is different for these two ribbon orientations (Supplementary Fig. 8).

The evolution of the HPhP resonances with changing $w$ can be understood by considering that the excitation of HPhPs are originated from the synergy between guided waves of the array and FPRs in an individual ribbon: the scattering of light at the ribbon edges excite the polariton FPR, while the guided waves further couple with and transfer energy to the polariton waves (see Supplementary Note 2 and Supplementary Fig. 5, 6). The conditions for the occurrence of the FPRs and guided waves are $q_{PhPs}w = \pm m\pi$ and $q_{PhPs}\Lambda = \pm n2\pi$, respectively,

with $m, n = 1, 2, 3$,[45,57]. Because in our study, the $\Lambda$ is deliberately set as $2w$, these two equations are the same. Therefore, each $w$ corresponds to an in-plane polariton wavevector of $q_{PhPs} = \mp \frac{m\pi}{w}$. When $w$ changes the resonance peak will scale according to the in-plane polariton dispersion relations, $\omega(q_{PhPs})$. This can be readily seen by calculating the $\omega(q_{PhPs})$ along the [100] and [001] directions, which is visualized as the 2D false color plot of the imaginary part of the complex reflectivity $\text{Im}r_p(q_{PhPs}, \omega)$ (see Supplementary Note 4 for details on the calculation of $\text{Im}r_p$). The PhP resonance frequencies obtained from the spectra shown in Fig. 2a, b are then overlaid onto the same plot by using $q_{PhPs} = m\pi/w$. Excellent agreement is obtained between the experimental measurements and calculated lowest-order ($m = 1$) HPhP branches for all three bands (Fig. 2c). Such an agreement further validates that the highly anisotropic HPhPs in the α-MoO₃ flake are directly excited from the far-field with the help of the in-plane hyperbolic polariton tuners. It is noted that the reflection of polariton waves by the ribbon edges may induce phase shifts, which can violate the condition for the FPRs by a phase of $2\Phi$[58]. A previous theoretical result showed that in monolayer graphene, the $\Phi$ for plasmon polaritons is ~ 0.75π[58]. In comparison with the plasmon polariton in graphene, the propagation and reflection of the in-plane HPhPs are rather complicated, making the phase shift difficult to be predicted. In our analyses, the $\Phi$ is taken as π according to a very recent study[59]. The good agreement between the experimental measurements and calculated results further validates our setting.

The far-field reflectance spectra allow for evaluating the Q-factor of the HPhP resonance, which is defined as $Q = \frac{\omega_0}{\Gamma}$, with $\omega_0$ and $\Gamma$, the frequency and linewidth of a specific resonance[7] (see details in Supplementary Note 1 for extraction of the Q-factors). For HPhP resonances in Band 1 and 3, their Q-factors respectively decrease and increase monotonically against $\omega_0$ (Fig. 2d). This is because when $\omega_0$ increases, the PhP resonance in Band 1 shifts closer to the $\omega_{TO}^{100}$ (820 cm⁻¹), while the resonance in Band 3 shifts away from the TO phonon mode along [010] axis at 958 cm⁻¹ ($\omega_{TO}^{010}$)[10,36]. Thus the polariton dissipation by lattice absorption will be strengthened (suppressed) for Band 1 (Band 3), giving rise to a larger (smaller) $\Gamma$. The non-monotonic behavior of the Q-factor for the HPhP resonance observed in Band 2 can be understood by considering that, in addition to $\omega_{TO}^{100}$, there is another LO phonon mode along the [100] axis at 972 cm⁻¹ ($\omega_{LO}^{100}$)[10,36]. Leaky-mode absorption induced by ENZ condition also occurs near the $\omega_{LO}^{100}$. Therefore, the Q-factor first increases as $\omega_0$ is farther from the $\omega_{TO}^{100}$, and then decreases gradually approaching the $\omega_{LO}^{100}$. It is noted that the Q-factors observed in Band 1, 2, and 3 are 15−25, 25−100, and 200−300, respectively. Most of these values are higher than those observed in graphene nano-gratings with similar resonance frequencies[41,42,47]. In particular, the Q-factor of the Band 3 resonance can be as high as 300, which is on par with the highest observed in hBN nanoresonators (360) via the same far-field technique[60]. Such high Q-factors, coupled with the small modal volumes and footprint of the 1D-PRPs, indicate that the α-MoO₃ tuners offer important application potential for high-efficiency compact photonic devices and components, as demonstrated below.

The far-field spectra also allow for extracting the polariton lifetimes, which span from 0.2 to 3.0 ps in the three Reststrahlen bands

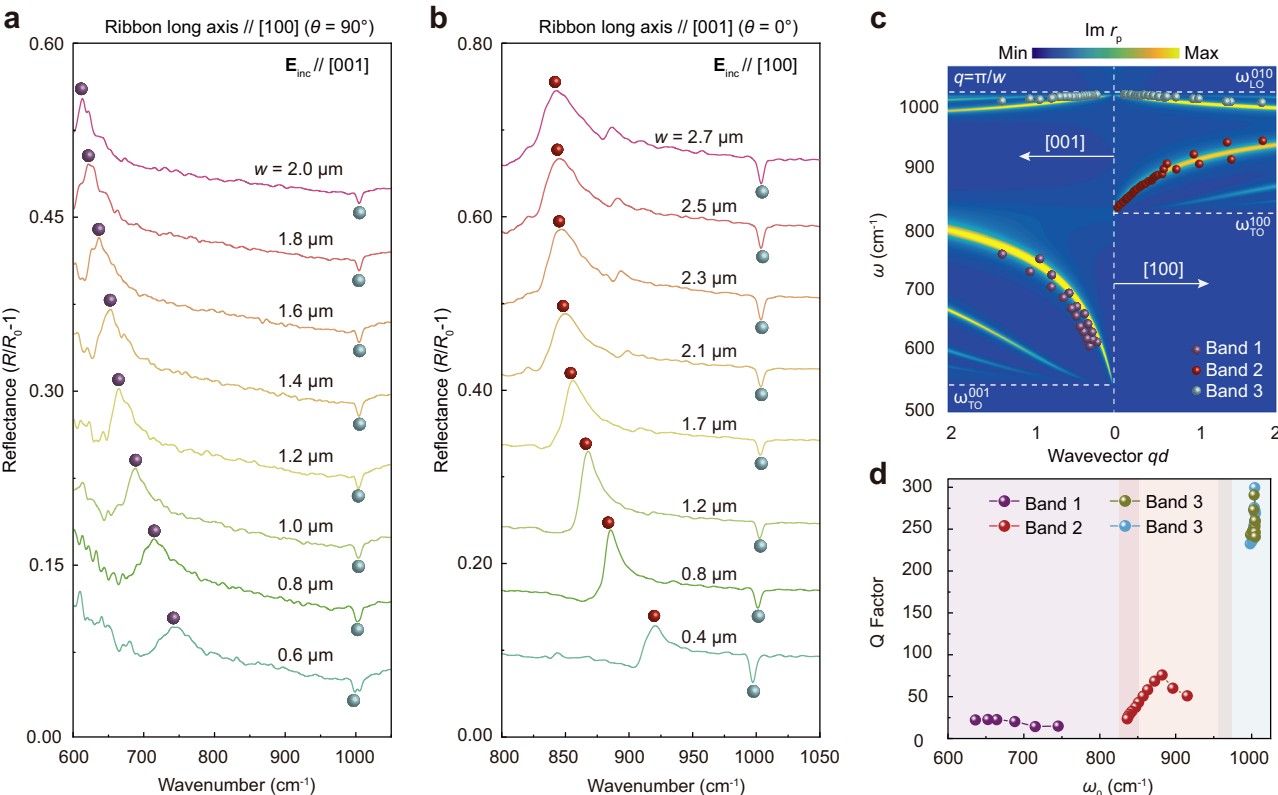

**Fig. 2 | Tuning HPhP resonances with tuner patterns of different ribbon widths. a, b** Experimentally measured reflectance spectra of tuners with different ribbon widths $w$, showing the large resonance shift in the reflectance spectra by changing $w$ within one micrometer. The ribbon long axes are paralleled to the [100] (**a**) and [001] (**b**) crystallographic directions, respectively. The colored spheres indicate the resonance frequencies. **c** Comparison of the theoretically derived (false color plot) and experimental (color spheres) HPhP dispersion relations, showing the good matching between the two results. The color spheres are extracted from reflectance spectra at different $w$. The false color plot represents the calculated imaginary part of the complex reflectivity, $\text{Im} r_p(q_{\text{PhPs}}, \omega)$, of the air/α-MoO₃/SiO₂/Si multilayered structure. The polariton wavevector is normalized by the thickness $d$ of the α-MoO₃ flake. **d** Dependence of quality factor (Q-factor) on resonance frequency $\omega_0$ extracted from the reflectance spectra shown in (**a**) and (**b**). In Band 1 and 3, the Q-factors respectively decrease and increase monotonically as the $\omega_0$ increase, whereas the non-monotonically trend has been observed in Band 2. Colored shaded regions mark the frequency ranges of different Reststrahlen bands of α-MoO₃.

(Supplementary Note 5, Supplementary Fig. 9, and Supplementary Table 1). It is noted that the fabrication of 1D-PRPs can lead to damage to the ribbon edges, which can reduce the polariton lifetimes and Q factors. Usually, this issue is inevitable during the patterning of the vdW crystal for various characterizations and device applications. To evaluate the additional polariton damping induced by the patterning processes, we performed near-field measurements on the same α-MoO₃ flake in the unpatterned region and extracted the intrinsic polariton lifetimes (Supplementary Note 5, Supplementary Fig. 9, and Supplementary Table 1). In comparison with the unpatterned α-MoO₃, the phonon polariton lifetime in the ribbon arrays are reduced by 20−57%. The reduction in the lifetime of polariton modes in α-MoO₃ ribbons is attributed to defects or impurities at the rough edges of the ribbons introduced during the fabrication process. These imperfections can create additional scattering centers for HPhPs. Furthermore, they will also lead to the broadening of the pristine phonon modes. Raman spectroscopic characterizations show that, compared to the unpatterned region, the overall linewidths of the phonon modes in the ribbons increased by 4.6−14.1% (Supplementary Note 6, Supplementary Fig. 10, and Supplementary Table 2). This broadening shortens the lifetime of the associated polariton modes as well. Despite the lifetime reduction, the 1D-PRPs still exhibit high Q factors upto 300. Although such a value is smaller than that of resonances sustained by an unpatterned and naturally grown α-MoO₃ ribbon[61], it can be further improved by optimizing the processing parameters.

## Tuning the HPhPs with ribbon arrays of different skew angles

The in-plane HPhP dispersions of the α-MoO₃ are highly anisotropic[9,10]. This offers unique tunability of the HPhP resonances by changing the 1D-PRP orientations, which cannot be realized in vdw crystals with isotropic in-plane dispersions such as hBN and graphene. As such, 1D-PRPs with fixed $w$ (480 nm), but different $\theta$ were fabricated (Figs. 3a, 1c). Each pattern can provide polariton momenta of $q_{\text{PhPs}} = \pi/w$, with the direction perpendicular to the ribbon long axis. In this way, when the ribbon is rotated away from the [001] axis, HPhPs with wavevectors of different orientations within the basal plane can be excited, giving rise to HPhP resonances that are strongly dependent on the $\theta$. Specifically, HPhP resonances in Band 2 and 3 can be observed for $\theta = 0°$, and all of the HPhP resonances in Band 1, 2, and 3 appear when $\theta$ is increased (Fig. 3b). For $\theta = 90°$, the resonance in Band 2 merges with the $\omega_{\text{TO}}^{100}$. In addition, the resonances in Band 1 and 2 are highly sensitive to $\theta$, while that in Band 3 shifts slowly against $\theta$ (Fig. 3b). This can be ascribed to the distinct in-plane polariton dispersions of the three bands. Specifically, the in-plane isofrequency contour (IFC) of HPhPs in Band 3 is an ellipse, where the polariton dispersion differs moderately along different directions in the basal plane. However, the IFCs in Band 1 and 2 are hyperbola, making their polariton dispersions highly dispersive with $\theta$. These angle-dependent behaviors can be seen more clearly by plotting the dependence of resonance frequencies in the three bands on the skew angle (Fig. 3c), which agrees well with the calculated $\text{Im} r_p(q_{\text{PhPs}}, \omega)$.

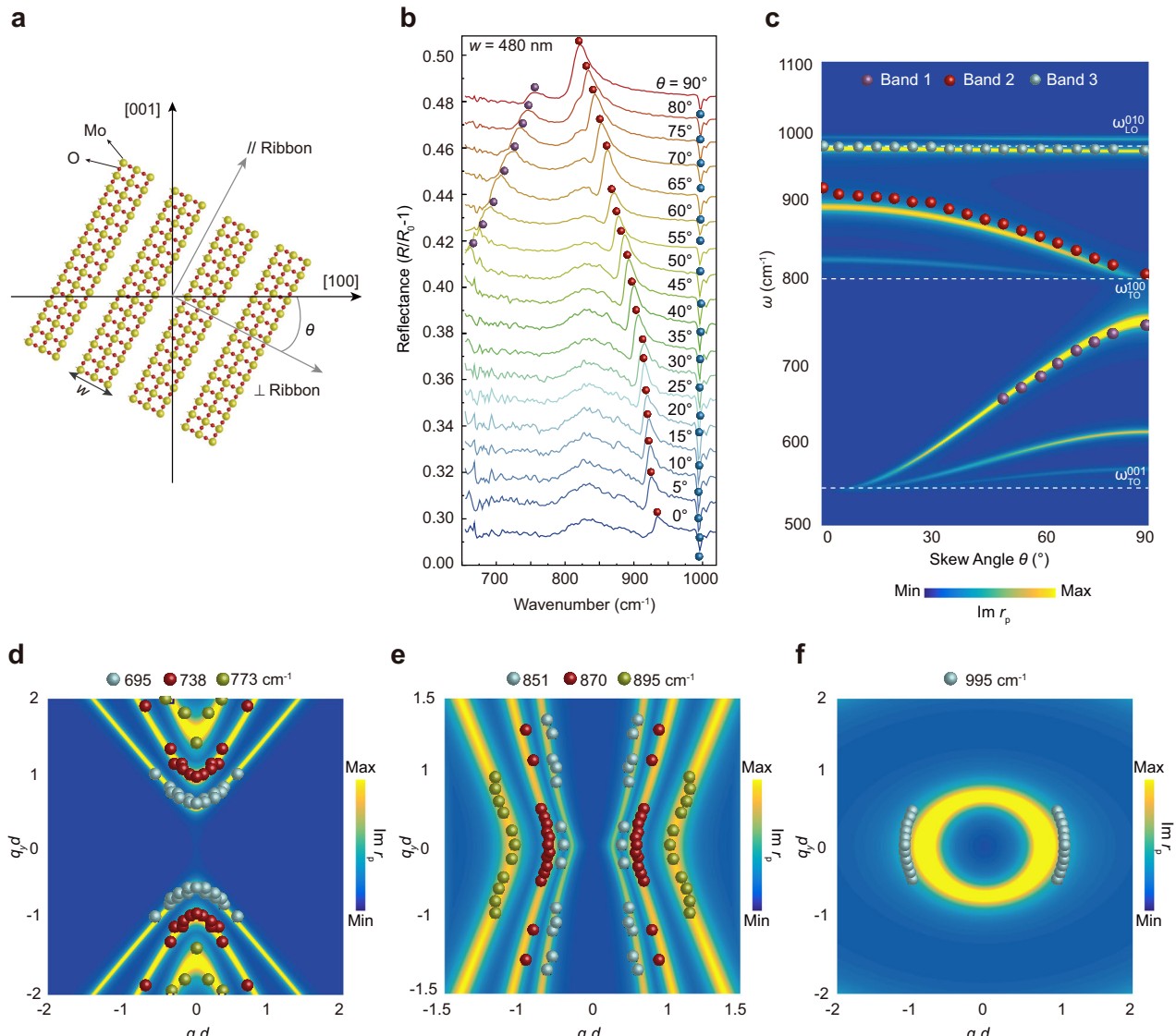

**Fig. 3 | Tuning the HPhPs with periodic tuner patterns of different skew angles.** **a** Schematic showing 1D periodic tuner patterns with a skew angle $\theta$. **b** Experimental reflectance spectra of 1D periodic tuner patterns with different $\theta$, showing that when $\theta$ increases, the resonance in Band 1 clearly blueshifts, whereas a reverse trend appears in Band 2, and a slow variation of the resonance occurs in Band 3. The widths of the ribbons are kept at $w = 480$ nm. The colored spheres depict the resonance frequencies. **c** HPhPs resonance frequency, $\omega_O$, as a function of $\theta$, showing the consistence between the calculation and experimental results. The false color plot represents the calculated $\text{Im}r_p(\theta, \omega)$ of the air/$\alpha$-MoO$_3$/SiO$_2$/Si multilayered structure. The colored spheres correspond to the experimental

resonance peaks extracted from the curves in (**b**). **d**–**f** Polariton isofrequency contours (IFCs) at different frequencies in Band 1 (**d**), Band 2 (**e**), and Band 3 (**f**), respectively, showing clear hyperbola for Band 1 and 2, while an ellipse for Band 3. The false color plots represent the $\text{Im}r_p(\omega, q_{\text{PhPs}x}, q_{\text{PhPs}y})$. The colored spheres in the first quadrant represent the experimental resonance peaks at $695.0 \pm 0.9$, $738.0 \pm 0.8$, $773.0 \pm 0.7$, $851.0 \pm 0.9$, $870.0 \pm 0.9$, and $895.0 \pm 0.9$ cm$^{-1}$. Spheres in other quadrants are duplicated according to the symmetry of the measurement scheme and the $\alpha$-MoO$_3$ crystal. The wavevectors in (**c**–**f**) are normalized by the thickness $d$ of the $\alpha$-MoO$_3$ flake. Parameters $q_{\text{PhPs}x}$ and $q_{\text{PhPs}y}$ represent the $x$ and $y$ components of the polariton wavevector, respectively.

The HPhP dispersion relations at each skew angle can be obtained by measuring the reflectance spectra from 1D-PRPs with different $w$ ($q_{\text{PhPs}}$) at a specific $\theta$ (Supplementary Fig. 12), whereby the in-plane polariton IFCs at different energies can be re-constructed and visualized. Clearly, the HPhP resonances in Band 1 and 2 depict IFCs of open hyperbolic shapes (Fig. 3d, e), while those in Band 3 correspond to an IFC of a closed ellipse (Fig. 3f). Moreover, at higher frequencies in Band 1 (Band 2), the opening-angles of the hyperbolic sectors become smaller and the hyperbola bends toward the [001] ([100]) direction (Fig. 3d, e). All the experimental points can be fit well by the calculated $\text{Im}r_p$ (pseudo-colored plots shown in Fig. 3d–f and Supplementary Fig. 12). These results provide further direct evidence for far-field excitation and modulation of the hyperbolic HPhPs in the $\alpha$-MoO$_3$ flake.

High-Q resonances can also be induced in 1D-PRPs made out of an in-plane isotropic polaritonic vdW crystal[44]. The frequency of such resonances can also be tuned by changing the ribbon width, but the $\alpha$-MoO$_3$ 1D-PRPs proposed in the current study is unique. The in-plane hyperbolicity of $\alpha$-MoO$_3$ makes it possible to tune the resonance frequency of the 1D-PRPs by rotating the ribbons while fixing their widths. This feature can greatly simplify the fabrication processes of arrays with different resonance frequencies, and small additional damping will be introduced because the ribbon width is unchanged. Moreover, for a polariton mode with a wavevector approaching the asymptote of the hyperbolic IFC, the polariton momentum will become remarkably high. This will generate much stronger electromagnetic field confinement than those with wavevectors away from the asymptote. These

modes can significantly enhance light–matter interactions at the nanoscale and lead to various applications, such as enhanced light emission, ultrasensitive biosensing, and nonlinear optical signal generations. For these applications, usually, a fixed operation frequency is preferred. In the 1D-PRPs made out of α-MoO$_3$, tuning the orientation and width of the ribbons can both tailor the resonance frequency. Therefore, it is possible to induce a high-momentum polariton mode while fixing its resonance frequency by simultaneously orienting the ribbon long axis along the asymptote of the hyperbolic IFC and tuning the ribbon width. This feature can open an avenue for the applications just mentioned.

## Far-field excitation of tunable THz HPhPs with ribbon arrays

The α-MoO$_3$ flake can also support nanoscale-confined HPhPs in the THz domain from 260–400 cm$^{-1}$ (8–12 THz), which, however, have only been probed using the near-field nano-imaging technique[30]. To demonstrate the far-field excitation and tuning of the THz HPhPs, we fabricated 1D-PRPs of different $w$ and $\theta$ and characterized their spectral responses. Due to the relatively low signal-to-noise ratio of the bolometer in the THz domain, transmission spectra were recorded, which is defined as $T/T_0$, with $T$ and $T_0$ the transmittance of the light through the sample and bare substrate (Methods). For 1D-PRPs with long axes parallel with [100] and [001] directions, we observed that resonance valleys associated with polariton wavevectors along the [001] (HPhP$_{[001]}$, left panel) and [100] (HPhP$_{[100]}$, right panel) direction can be excited in the spectral range of 270–330 cm$^{-1}$ and 350–400 cm$^{-1}$, respectively (Fig. 4a). The tunability of these two resonances is clearly demonstrated by their redshifting behaviors with increasing $w$. The polariton dispersion relations were then obtained by extracting the resonance frequencies at different $w$. The analytical dispersions were derived using the dielectric tensor reported in ref. 30 and are in good agreement with the experiment measurements (Fig. 4b and Supplementary Note 7). The Q-factors of the two types of resonances can be evaluated according to the transmittance spectra shown in Fig. 4a, which both increase with increasing the resonance frequency (Fig. 4d). The available Q-factors are in the range of 15–25 (HPhP$_{[001]}$) and 23–47 (HPhP$_{[100]}$) respectively, which surpass that of graphene plasmon resonance in the THz regime[40].

The HPhP resonances are also strongly dependent on the skew angle of the 1D tuner patterns. By sweeping the $\theta$ from 0° to 90° (ribbon long axis rotating from [001] to [100] direction), the HPhP$_{[001]}$ and HPhP$_{[100]}$ resonances shift monotonically to higher and lower frequencies, respectively (Fig. 4c). Moreover, with the experimental data shown in Fig. 4b, c, the in-plane IFC of the HPhP$_{[001]}$ can readily be drawn to exhibit a clear hyperbola opening towards the [001] axis (Supplementary Fig. 13). These results corroborate the excitation of HPhP resonances in THz Reststrahlen Band 1 (HPhP$_{[001]}$) and 3 (HPhP$_{[100]}$), which are consistent with the previous nano-imaging results[30].

## Tunable LWIR and THz polarization notch filters

A PNF is a unique optical component that combines a polarizer and a narrow band-rejection filter together into a single component, which is able to block a monochromic laser with a given linear polarization, while passing light of all polarization states at wavelengths adjacent to the laser line. Such a filter has broad application prospects in laser spectroscopy and optical communications, but the commercial products are rare, and especially, there is no commercial PNF in the LWIR and THz ranges. The salient high-Q (Figs. 2d, 4d) and polarization-sensitive (Fig. 1e, f) α-MoO$_3$ tuners established above provide opportunities for developing tunable PNFs[62]. As such, we constructed PNFs using 1D-PRPs with long axes along [001] and [100] axes, respectively. For typical 1D-PRPs with $w = 1000$ nm, their extinction spectra in LWIR (Fig. 5a) and THz (Fig. 5b) regimes are strongly polarization-dependent. Specifically, the HPhP resonances only appear when the

excitation polarization is perpendicular to the ribbon's long axis. This can be seen more clearly by plotting the extinction at the corresponding resonance peaks, i.e., 650 cm$^{-1}$/270 cm$^{-1}$ and 869 cm$^{-1}$/362 cm$^{-1}$, against light polarization for ribbons parallel to [100] and [001] directions, respectively (Fig. 5c, d). The performance of a PNF for blocking a monochromatic light source can be evaluated by two parameters: the polarization extinction ratio and bandwidth. Specifically, the polarization extinction ratio is defined as $10 \log(T_0/T)$, with $T_0$ and $T$ the transmittance of light polarized along and perpendicular to the ribbon's long axis at the resonance frequency. The bandwidth can be calculated according to the full width at half maximum (FWHM) of a specific resonance. Accordingly, for the PNFs with resonances at 650, 869, 270, and 362 cm$^{-1}$, corresponding extinction ratios (bandwidth) are 3.4 dB (41.5 cm$^{-1}$), 5.5 dB (26.5 cm$^{-1}$), 4.5 dB (23.4 cm$^{-1}$), and 3.7 dB (24.5 cm$^{-1}$), respectively. In particular, the bandwidths of the PNFs are comparable to many of the commercial narrow-band-pass filters in similar spectral regimes, which are usually polarization insensitive, (Supplementary Fig. 14), while the thicknesses of the PNFs (200 nm) are much smaller than those of the commercial components (~1 mm). More importantly, using 1D-PRPs with different $w$, the operational frequency, polarization extinction ratio, and FWHM of the PNF can be engineered continuously (Supplementary Figs. 15, 16). The maximum peak extinction can reach 6.5 dB and the smallest FWHM can be as narrow as 17 cm$^{-1}$.

## Discussion

We have successfully demonstrated direct far-field excitation and characterization of the tunable LWIR and THz HPhPs in biaxial vdW α-MoO$_3$ patterned into simple 1D-PRPs. The 1D-PRPs can act as polariton tuners that are sensitive to the excitation polarization and with light extinction ratios up to 6.5 dB and high Q-factors up to 300. Such a compositional set of output functions are tunable and strongly dependent upon the in-plane hyperbolic phonon polaritons in the α-MoO$_3$. It is noted that in comparison with the recently reported patterned vdW WTe$_2$ flakes with tunable in-plane hyperbolic plasmon polaritons at cryogenic conditions[45], the polariton tuners proposed in the current study may be more favorable for practical applications because of their room-temperature operation, broader spectral range, and much higher quality factors of the resonance modes.

On the prospects, these in-plane hyperbolic polariton tuners can be used in optical circuitry, instruments and even modern information systems. The in-plane hyperbolic polariton tuner also opens up avenues for a variety of practical photonic and optoelectronic applications besides the PNFs shown in Fig. 5. For example, by engineering the tuner to spectrally overlap the HPhP resonance with a specific vibration or rotation transition of a molecule, strong interactions between the tuner and molecule can be induced[44,63], which can significantly enhance the molecular absorption or emission and give rise to various ultrasensitive bio-sensing techniques. The tuners with high-Q and tunable HPhP resonances can also be employed to regulate the blackbody emission[62], whereby narrow-band, polarized, and tunable thermal emission can be achieved[64–66]. Moreover, tunable and high-performance LWIR and THz photodetectors can also be envisioned by taking advantage of the strong light field localizations (Supplementary Fig. 7) and semiconductor nature of the α-MoO$_3$[19]; this type of devices may give unique functions necessary for future communication and radar applications.

For fundamental research, the far-field excitation methodology can complement near-field nano-imaging techniques and make the characterizations of PhPs of materials more precisely, especially for those with in-plane hyperbolicity. For example, in nano-imaging measurement, because the polariton waves are launched by an antenna (the scanning tip or antenna fabricated onto the sample surface), in principle, these HPhPs can propagate along different directions determined by the hyperbolic IFC[8–10]. Therefore, the measurements of

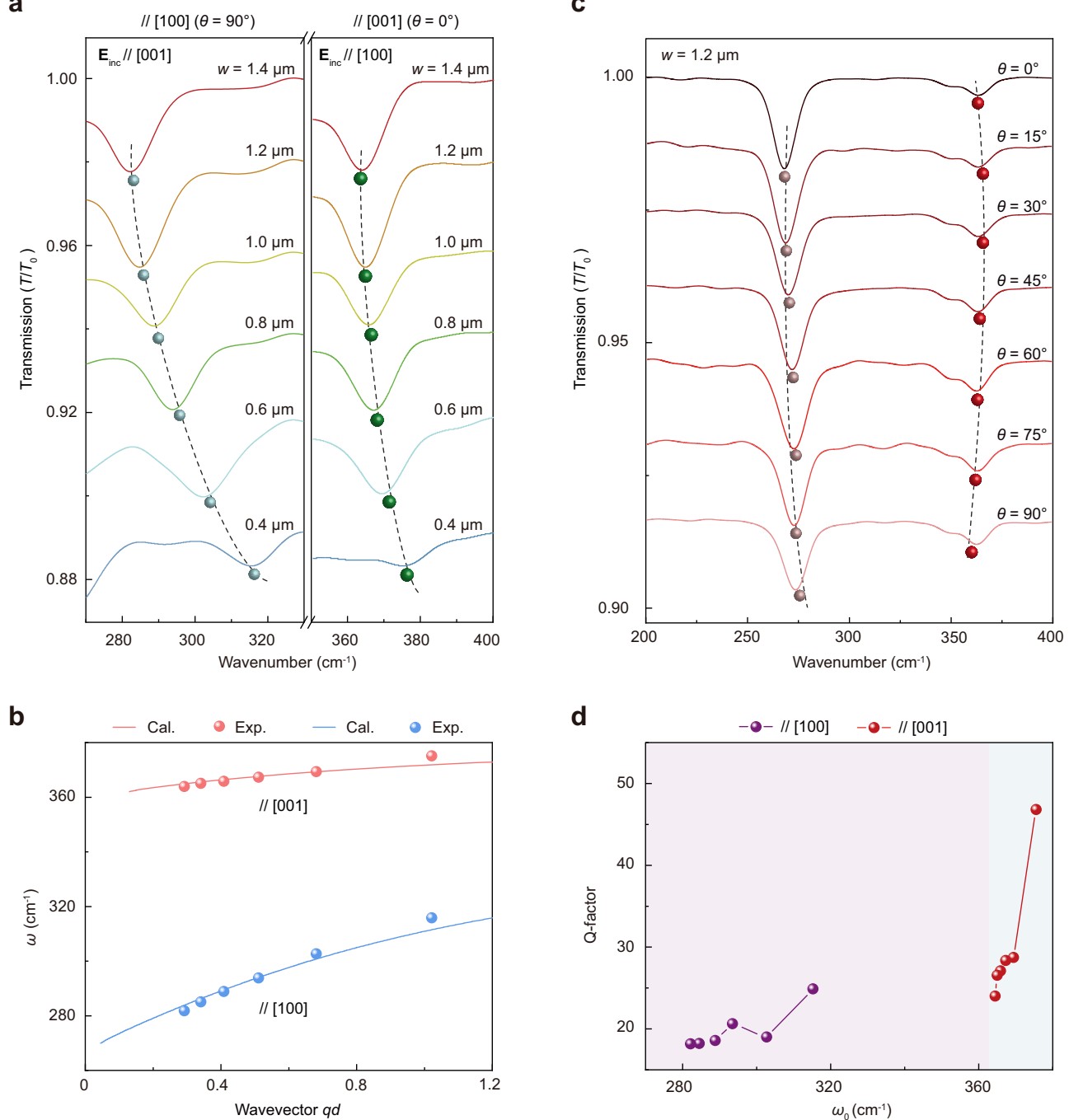

**Fig. 4 | Far-field excitations of tunable THz HPhPs in α-MoO₃ one-dimensional periodic tuner patterns. a** Experimental transmittance spectra of tuner patterns with different ribbon widths, showing redshifting behaviors of the two resonance valleys with changing $w$. The long axes of the ribbons are paralleled to the [100] (left panel) and [001] (right panel) crystallographic directions, respectively. The colored spheres indicate the resonance frequencies. The dashed lines are guides to the eye. **b** HPhP dispersion relations in the THz regime, showing the consistency between the results of calculation and experiment. The wavevector is normalized by the thickness of the α-MoO₃ flake. The colored solid lines are the calculated results according to the analytical dispersion relations of HPhPs propagating in the α-

MoO₃ flake. The colored spheres show the experimental results extracted from (**a**). **c** Experimental transmittance spectra of tuner patterns with different $\theta$, showing that as $\theta$ increase, the HPhP$_{[001]}$ and HPhP$_{[100]}$ resonances shift monotonically to higher and lower frequencies, respectively. The ribbon widths are fixed as $w = 1200$ nm. The colored spheres indicate the resonance frequencies. The dashed lines are guides to the eye. **d** Dependence of Q-factor on resonance frequency $\omega_0$, showing that in Band 1 and 3, the Q-factors increase monotonically as the $\omega_0$ increases. The solid spheres are extracted from the transmittance spectra in (**a**). The solid lines are a guide for eyes. Colored shaded regions in (**d**) mark the frequency ranges of different Reststrahlen bands of α-MoO₃.

the polariton wavelength and dispersion relation from the polariton interference fringes can be disturbed by these various polariton waves. On the other hand, with far-field excitation, the polariton wavevector is determined by the tuner's structural parameter and orientation. For a specific pattern, only one HPhP mode can be excited, which allows for

a more accurate characterization of the intrinsic HPhP properties. Additionally, it is possible to characterize the HPhPs using a broadband light source covering a broad THz spectral range. This can unveil more complete polaritonic properties for broadening and deepening our understanding of the THz HPhPs, in particular for two-dimensional

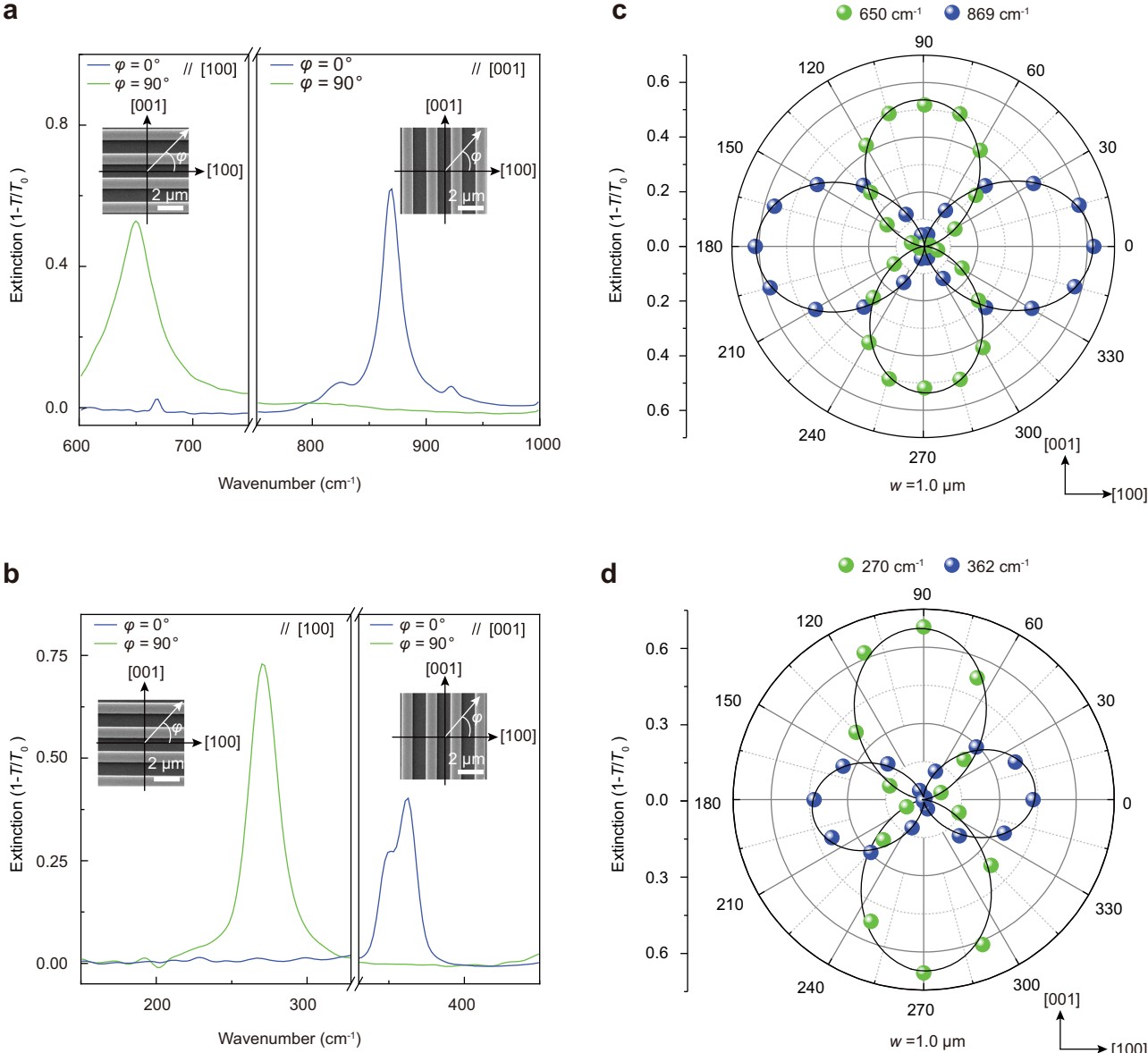

**Fig. 5 | Tunable polarization notch filters in THz and LWIR regimes made up of in-plane hyperbolic polariton tuners. a, b** Polarized extinction spectra of LWIR (**a**) and THz (**b**) notch filters. The long axes of the ribbons are parallel to [100] (left panels) and [001] (right panels) crystallographic directions, respectively. The widths of the ribbons are 1.0 μm. Insets: Schematic of the polarization excitation. $\varphi$ is the angle between the incident electric field and [100] crystallographic direction. The results show that the polariton resonances only appear when the excitation polarization is perpendicular to the ribbon's long axis. **c, d** Polar plots of the extinction as a function of excitation polarization at resonances frequencies of 650 cm⁻¹/869 cm⁻¹ (**c**) and 270 cm⁻¹/362 cm⁻¹ (**d**). The symbols and solid lines are experimental data and the corresponding fittings using a cosine squared function. The long axes of the ribbons are parallel to [100] (corresponding to resonances at 650 and 270 cm⁻¹) and [001] (corresponding to resonances at 869 and 362 cm⁻¹) crystallographic directions, respectively.

atomic crystals (e.g., the in-plane IFCs in the THz domain as shown in Supplementary Fig. 13), which is now limited by the discrete laser lines used in most nano-imaging measurements[30].

It is noted that in the current study, as a demonstration of principle, we only employ the simplest form of patterns, and only demonstrate with one type of material. In fact, patterns can be chosen depending on desirable applications, and also materials as long as supporting HPhPs in the THz and LWIR regimes[6,67]. The tuner allows us to modulate the wavefront of the incident light and control their power flow in an engineered space[21–26], which therefore enables a variety of interesting applications, such as negative refraction, holography, metalens, polarization conversion, and even topological PhPs and exciton-polaritons with robust beam steering functionalities[68,69], in

visible and near-infrared ranges to be expanded into the THz and LWIR spectral regimes.

When preparing the revised manuscript, we became aware of a recent work[59] on a similar topic to our current study.

## Methods

### Synthesis of large-area vdW α-MoO₃ flake

vdW α-MoO₃ thin flakes with a centimeter lateral size were prepared using a modified thermal physical vapor deposition method. Specifically, an alumina crucible filled with 0.1-g α-MoO₃ powders was placed at the center of a quartz tube as the evaporation source. Another crucible covered with a silicon substrate of 1 cm × 1 cm was placed 20 cm away from the source. The source was then heated up to 780 °C

and held at that temperature for 2 h. The α-MoO$_3$ powders were sublimated and crystallized onto the silicon wafer. Afterward, the quartz tube was cooled down to room temperature naturally. The large-area α-MoO$_3$ flakes can be found on the silicon wafer.

### Fabrication of the vdW α-MoO$_3$ 1D periodic tuner patterns

The as-grown α-MoO$_3$ flakes were transferred to a pristine (highly resistive) silicon substrate covered with a 300-nm oxide layer. Afterward, a selected flake was patterned into 1D-PRPs with different $w$ and $\theta$ using a combination of electron beam lithography (EBL: EBPG5000+, Netherlands) and reactive ion etching (RIE: 50 W for 10 min). For the EBL processing, a 400-nm layer of Poly(methylmethacrylate) (PMMA) photoresist was used. For the RIE etching, a mixture of O$_2$ (12 vol.%), Ar (30 vol.%), and CHF$_3$ (58 vol.%) was employed. The etching was conducted at 50 W for 10 min. To guarantee good signal-to-noise ratios of the spectral characterizations, the areas of the patterns were set as 50 μm × 50 μm and 300 μm × 300 μm for LWIR and THz regimes, respectively.

### Reflectance and transmittance spectral characterizations

LWIR and THz spectral characterizations were performed using a Bruker FTIR spectrometer (Vertex 70 v) integrated with a Hyperion 3000 microscope and a mercury cadmium telluride (HgCdTe) photoconductor (for the measurement of LWIR spectra) or a liquid–helium-cooled silicon bolometer (or the measurements of THz spectra) as the detector. A broadband black-body light source covering the LWIR and THz spectral regimes was employed as the incident light. A 15× reflective Schwarzschild objective was utilized to focus the incident light onto the tuner patterns with a spot size of ~500 μm. The reflected or transmitted light were collected from an area of 50 μm × 50 μm and 300 μm × 300 μm for LWIR and THz regimes, respectively, with the help of an iris diaphragm. For the polarization-dependent measurements, a polarizer was used to control the polarization of the incident light. For the reflectance and transmittance spectra of the tuner, a bare silicon substrate is used as reference for normalization. To characterize the PNFs in LWIR and THz regimes, a linear polarizer was placed before the detector to determine the polarization state of light transmitting through the tuners upon an unpolarized illumination.

## Data availability

Relevant data supporting the key findings of this study are available within the article and the Supplementary Information file. All raw data generated during the current study are provided in the Source Data file or available from the corresponding authors upon request. Source data are provided with this paper.

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

## Acknowledgements

H.C. and S.D. acknowledge support from the National Key Basic Research Program of China (grant nos. 2019YFA0210200 and 2019YFA0210203), H.C. acknowledges support from the National Natural Science Foundation of China (grant no. 91963205), the Guangdong Basic and Applied Basic Research Foundation (grant no. 2020A1515011329), and the Changjiang Young Scholar Program. Z.Z. acknowledges the support from the National Natural Science Foundation of China (grant no. 11904420), the Guangdong Basic and Applied Basic Research Foundation (grant no. 2019A1515011355), and the project funded by China Postdoctoral Science Foundation (grant no. 2019M663199). H.G.Y. acknowledges support from the National Natural Science Foundation of China (grant no. 12074085). J.D.C. acknowledges support from National Science Foundation #2128240. T.G.F acknowledges support from University of Iowa startup funding.

## Author contributions

H.C., S.D., N.X., and J.D.C. conceived and designed the experiments. W.H. fabricated the tuner patterns, characterized the far-field spectra, and analyzed the data. T.G.F., Q.X., and Z.Z. participated in the far-field spectroscopy measurements. T.G.F., H.Y., and Q.X. measured the THz spectra. F.S and Z.Z. conducted the numerical simulations and theoretical calculations. J.J. helped fabricate the tuner patterns. H.C., S.D., N.X., J.D.C., and H.Y. coordinated and supervised the work and discussed and interpreted the results. H.C. and W.H. co-wrote the manuscript with the input of all other co-authors. W.H., T.G.F., F.S., and Z.Z. contributed equally to work.

## Competing interests

The authors declare no competing interests.
