## [Peer Review File · Nature Communications]

In-plane hyperbolic polariton tuners in terahertz and long-wave infrared regimesREVIEWER COMMENTS

Reviewer #1 (Remarks to the Author):

Huang et al investigate the IR transmission and reflection spectra of nanopatterned α -MoO₃. Here, the authors patterned α -MoO₃ flakes into an array of nanoribbons to open a radiative coupling channel for the phonon-polariton modes and analyzed the anisotropic resonant feature in far-field. I find that the experiment and theoretical analysis presented in this work are sound. However, I feel that this work is rather straightforward and an expected extension of the previous works; The anisotropic phonon-polaritons in α -MoO₃ have been extensively studied since the first near-field study by Ma et al. was reported [Nature 562, 557 (2018)]. It is also well known that one can induce polaritonic resonances with far-field excitation by patterning a polaritonic material into an array of nanoresonators [Nat Photonics 7, 394 (2013), Nano Lett 13, 2541 (2013)]. Note that, however, nanopatterning of van der Waals crystals and measuring the collective response of the numerous resonators in the far-field introduce undesirable artifacts such as rough edges and inhomogeneous broadening. Therefore, for fundamental characterization, a near-field study on an unpatterned crystal is preferred over the far-field measurement on an array of many resonators. Besides, polaritonic resonators based on α -MoO₃ have been recently studied and the reported Q factors are above 700, which is more than two times the value reported in this paper [ACS Nano 16, 3027 (2022)]. Hence, I think that this manuscript does not seem to meet the impact and novelty standard of Nature Communications, but would be suitable for a more specialized journal.

Reviewer #2 (Remarks to the Author):

Wuchao Huang and co-authors reported a far-field excitation of in-plane hyperbolic polaritons in α -MoO₃ crystals by structuring the medium with momentum-matched periodic grating patterns. The authors demonstrated that the device works as an in-plane hyperbolic phonon-polariton tuner and exhibits a unique set of characteristic functions such as polarizer and notch filter for LWIR and THz regimes. This is the first demonstration of far-field control of high-k in-plane hyperbolic phonon polaritons in the LWIR and THz regimes. I believe this study will drive the field of photonics and the advancement of 2D materials for optoelectronics in LWIR and the THz region. The analytical and numerical simulations well supported the experimental findings. The results are convincing, timely, and essential to the field of mid-infrared photonics and 2D materials. I believe the manuscript is worthy of publication in Nature Communications, provided a suitable (but minor) revision is made to support the authors' claims. The list of my questions/suggestions is provided below.

1. I believe the authors' reported study of in-plane hyperbolic polariton tuner is conceptually very similar to the earlier theoretical and experimental demonstrations of "Hyper Crystals" derived from structured plasmonics hyperbolic metamaterials (Narimanov, Evgenii E. "Photonic hypercrystals." Physical Review X 4.4 (2014): 041014., Galfsky, Tal, et al. "Photonic hypercrystals for control of light-matter interactions." Proceedings of the National Academy of Sciences 114.20 (2017): 5125-5129.). Authors may cite these first demonstration papers.
2. The authors patterned the α -MoO₃ with periodic grating patterns for far-field excitation of high momentum in-plane hyperbolic phonon-polaritons. These hyperbolic phonon-polaritons can be both surface states and bulk states. The near-field technique (NSOM) only excites and images the surface phonon-polaritons. However, in the current study, the outcoupled modes' character is not well-defined whether they are surface or bulk modes.
3. The experiments were performed using a commercially available Schwarzschild objective in the FTIR equipment. The NA of this objective is usually 0.58, implying that measured reflections correspond to averaged responses over +/- 350. At the same time, the reflection response from the analytical simulations was performed for each angle. I believe there should be a discrepancy in the theory and experimental results matching. However, the authors claim a good agreement between the theory and experiments. Can the authors explain the reason for no mismatch?
4. The authors mentioned they determined the bulk phonon modes of MoO₃ from Raman scattering measurements. Do these in-plane hyperbolic phonon-polariton modes also show the Raman scattering signal? I believe the authors should report these measurements to claim the phononic nature of these modes.

5. In Fig.1c, the authors show the SEM images of the patterns drawn at different skew angles. The pattern image for 300 skew angle seems to correspond to 600 and vice versa. I believe it is only a mistake, and actual measurements reported in Fig. 3 were not performed following this notation.
6. I suggest the authors cite the recent developments of phonon and exciton-polaritons in 2D materials, especially the topological phonon-polaritons (Guddala, S. et al. "Topological phonon-polariton funneling in midinfrared metasurfaces." *Science* 374.6564 (2021): 225-227.), and topological exciton-polaritons (Li, Mengyao, et al. "Experimental observation of topological Z2 exciton-polaritons in transition metal dichalcogenide monolayers." *Nature communications* 12.1 (2021): 1-10.).
7. I suggest the authors increase the text's font in all figures.
8. The authors should review Fig. S3. The THz region Reststrahlen bands were labeled reverse to the LWIR region bands.

Reviewer #3 (Remarks to the Author):

The authors report far-field (far-)infrared spectroscopy of ribbon arrays fabricated from large MoO₃ flakes. Results seem interesting and in some aspect new, but reading the manuscript the reviewer was sometimes confused by the description of the work. The authors do more or less the same as many others were doing before: fabricating ribbons from van der Waals materials/2D material/graphene/etc and then performing far-field infrared and THz spectroscopy. But the authors stress a certain challenge in doing so (although it is quite standard), and much more confusing, they introduce the expression "polariton tuner" which the reviewer does not understand well. The authors do very similar experiments as reported e.g. in ref. 50, just probing phonon polaritons instead of plasmon polaritons. There are also several scientific/technical details that need clarification. Details see below.

In summary, although the work principally is quite interesting, I find the wording and description of the work sometimes misleading/confusing for the reader. It may become suitable for *Nature Communications* when the authors describe their work and results more clearly regarding technical, scientific and novelty aspects, avoiding expressions that are not well defined/explained or justified (such as lattice, tuners, etc).

- 1) For example, the authors write the following without providing explanation or justification for the claims in the abstract:
 "Here, we propose an In-Plane Hyperbolic Polariton Tuner and demonstrate a first device responding to direct far-field excitation and giving rise to a unique set of characteristic functions at far-field."
 a) The reviewer does not see a device, just an array of ribbons.
 b) why "first" if same has been done in e.g. ref 50 and other works?
 c) what do the authors by mean by "set of functions at far field" when just recording far-field of static ribbons?
 d) what is unique regarding the functions and why, e.g. compared to ref 50?
- 2) The authors highlight advantages of ribbon spectroscopy compared to e.g. near-field microscopy studies. The discussion is not objective enough. For example, broadband infrared sources can be used and actually have been used. There are several papers published, e.g. ref. 5, 10, etc. Broadband spectra can be also obtained with EELS spectroscopy. The authors also do not consider that fabrication of ribbons may lead to damage of the ribbon edges that may give rise to additional damping and lowering of Q factors.
- 3) The authors introduce a reciprocal lattice momentum G , but what is meant by "lattice". Do the ribbon arrays have as a lattice, i.e. collectively (e.g. supporting lattice resonances) or do they couple to each other. It seems that they behave as individual ribbons and thus the words lattice, diffract, etc. are confusing/misleading. The authors also cite ref. 59 that is even more confusing because the phase matching condition is given by the period of the grating and not by the ribbon width.

4) The authors write that the polariton propagate along the ribbons and form standing-wave resonances due to reflection at ribbon edges. It seems that that the polaritons propagate transverse to the ribbons, i.e. across the ribbons rather than along the ribbons. I would say the authors measure Farby-Perot modes across the ribbons but they do not clearly explain this. The polaritons also may be phase shifted upon reflection at the edges, which needs to be considered in the analysis.

5) It is often not clear whether it is simply the high Q factor of the resonances or the anisotropy of the material that is important. For example, the authors mention tunable notch filters in Fig. 5. They show narrow resonances, which spectrally shift when the ribbon width is kept same but the orientation is rotated relative to crystal axes. This is interesting, but narrow resonances can be also shifted with ribbons made out of an isotropic material (where high Q factor polaritons exist) by changing the ribbon width. I am wondering what is the specific advantage of using anisotropic materials such as MoO₃, apart of the high Q factor that MoO₃ has (which is not necessarily a consequence of its anisotropy).

6) Since the authors claim that they report a first far field characterization, a more detailed discussion of the linewidths/lifetimes could be provided, particularly discussing whether fabrication uncertainties have some influence compared to lifetimes obtained from near-field characterization.

We thank the efforts and the valuable comments from the reviewers. We have done our best to carefully revised our manuscript according to the comments and suggestions of the reviewers. Our detailed response to the comments of the reviewers is included at the back of this letter. In addition, changes have been indicated in yellow in one of the uploaded manuscript files in the PDF format.

Response to Reviewer #1

We thank this reviewer for the rigorous and much valuable comments on our manuscript.

Comment: Huang et al investigate the IR transmission and reflection spectra of nanopatterned α -MoO₃. Here, the authors patterned α -MoO₃ flakes into an array of nanoribbons to open a radiative coupling channel for the phonon-polariton modes and analyzed the anisotropic resonant feature in far-field. I find that the experiment and theoretical analysis presented in this work are sound. However, I feel that this work is rather straightforward and an expected extension of the previous works; The anisotropic phonon-polaritons in α -MoO₃ have been extensively studied since the first near-field study by Ma et al. was reported [Nature 562, 557 (2018)]. It is also well known that one can induce polaritonic resonances with far-field excitation by patterning a polaritonic material into an array of nanoresonators [Nat Photonics 7, 394 (2013), Nano Lett 13, 2541 (2013)]. Note that, however, nanopatterning of van der Waals crystals and measuring the collective response of the numerous resonators in the far-field introduce undesirable artifacts such as rough edges and inhomogeneous broadening. Therefore, for fundamental characterization, a near-field study on an unpatterned crystal is preferred over the far-field measurement on an array of many resonators. Besides, polaritonic resonators based on α -MoO₃ have been recently studied and the reported Q factors are above 700, which is more than two times the value reported in this paper [ACS Nano 16, 3027 (2022)]. Hence, I think that this manuscript does not seem to meet the impact and novelty standard of Nature Communications, but would be suitable for a more specialized journal.

We thank very much the reviewer for raising important concerns on the advances of our current study in comparison with the previous reports. These concerns really help us improve the scientific quality of our study.

On the basis of our understanding, we summarize the reviewer's comments in four aspects, which we will provide point-to-point responses.

CI: I feel that this work is rather straightforward and an expected extension of the previous works; The anisotropic phonon-polaritons in α -MoO₃ have been extensively studied since the

first near-field study by Ma et al. was reported [Nature 562, 557 (2018)].

RC1: We agree with the reviewer that the infrared anisotropic phonon polaritons (PhPs) in van der Waals (vdW) α -MoO₃ have been extensively studied since the report in (Nature 2018, 562, 557. by Ma et al.), as well as the reports from our group that were published almost at the same time (Adv. Mater. 2018, 30, 1705318.; Sci. Adv. 2019, 5, eaav8690.). In these previous reports, the PhPs were characterized using the scattering-type scanning near-field optical microscope (s-SNOM). Due to the limited operation frequency of the laser source, the in-plan anisotropic dispersions of the PhPs were mostly demonstrated in the spectral regime from 820 to 1010 cm⁻¹, which only covers two of the three infrared Reststrahlen bands of the α -MoO₃ (Band 2 in the range of 820 – 972 cm⁻¹ and Band 3 in the range of 958 – 1004 cm⁻¹). It is therefore very difficult to investigate the polariton characteristics in Band 1, especially the in-plane isofrequency contour (IFC), *via* the near-field techniques. To the best of our knowledge, in the paper published in 2019 from our group (Sci. Adv. 2019, 5, eaav8690.), with the help of sophisticated photo-induced force microscopy (PiFM) technique, which is also a near-field technique, we reported the first experimental measurement to date on the hyperbolic PhPs (HPhPs) dispersions in Band 1 of α -MoO₃. However, only dispersions along two specific crystallographic directions, *i.e.*, [100] and [001] directions, were measured, leaving the IFC in this band unresolved. In the current study, by patterning the α -MoO₃ into one-dimensional (1D) ribbon arrays, we managed to address the in-plane polariton IFCs in all of the three bands, especially in the Reststrahlen Band 1 (545 – 851 cm⁻¹). This is because our measurements were conducted using a far-field spectroscopic technique, whereby the light source integrated in the Bruker FTIR spectrometer (Vertex 70v) operates in a broad spectral range covering the mid-infrared to THz regimes (100 – 4000 cm⁻¹). From the fundamental point of view, our results advance towards a further understanding of the HPhPs in the α -MoO₃ compared to the previous studies based on near-field measurements.

More importantly, the previous studies have only shown the exotic polaritonic properties of the α -MoO₃ flake, while to the best of our knowledge, a demonstration of these characteristics in a practical device application is not given. The challenge lies in that one needs to compensate the large photon–polariton momentum mismatch for **far-field excitation of the HPhPs**. The near-field excitation relies on utilizing a metallic nanotip to compensate the large momentum mismatch between free-space photons and polaritons, which can only provide fundamental characterization of the HPhPs. For most of practical device applications, a far-field excitation of the HPhPs is much more preferred. In our current study, to take a step forward, in addition to reveal the polaritonic characteristics of the α -MoO₃, we intend to **demonstrate the possible practical application of the in-plane anisotropic HPhPs in α -MoO₃**. Direct far-field excitation and characterization of the HPhPs can on one hand directly obtain parameters related to the photonic applications, such as polarization characteristics, optical filtering characteristics, Q-factors, *etc.*, and on the other hand directly demonstrate the device application, just as the polarizing notch filters (PNFs) shown by our study. The PNFs is a unique optical component that combines a polarizer and a narrow band-rejection filter together into a single component, which

is able to block a monochromatic laser with a given linear polarization, while passing light of all polarization states at wavelengths adjacent to the laser line. Such a filter has broad application prospects in laser spectroscopy and optical communications, but the commercial products are rare, and especially, there is no commercial PNF in the long-wave infrared (LWIR) and terahertz (THz) ranges.

On the basis of these considerations, we neither consider our study to be straightforward nor a simple extension of the previous works.

To strengthen the advance of our findings, a discussion has been added paragraph 3 in page 2 as,

"However, so far the demonstration of utilizing the above exotic characteristics in a practical device is not given. The challenge lies in that one needs to compensate the large photon–polariton momentum mismatch for far-field excitation and far-field characterization of the HPhPs. In the previous studies the HPhPs of vdW crystals have been observed by near-field nano-imaging techniques^{8–10,28}, relying on using a metallic nanotip to compensate the large momentum mismatch between free-space photons and polaritons. For most of practical device applications, direct excitation of the HPhPs from the far-field is necessary. Some earlier studies indicate that it is possible to pattern the surface of vdW crystals, such as with graphene^{40–42}, hexagonal boron nitride^{7,43,44}, semi-metals⁴⁵, and topological insulators⁴⁶ to excite and measure the various types of polaritons. However, these previous studies focused on plasmon polaritons with in-plane isotropic^{40–42,46} and hyperbolic dispersions⁴⁵, as well as phonon polaritons with in-plane isotropic dispersion^{7,43,44}. Far-field excitation and characterization of the tunable in-plane HPhPs in vdW crystals, especially in THz spectral regime, remain unexplored. Furthermore, exploring the applications of the in-plane HPhPs in optical devices has so far remained elusive. These can be done if one has a vdW crystal with large enough lateral size while maintaining thicknesses of nanometer scale, so that patterns are made larger than the diffraction limit for these free-space wavelengths and thus, suitable for far-field spectroscopy.

In this article we demonstrate for the first time far-field excitation and far-field characterization of HPhPs in an In-Plane Hyperbolic Polariton Tuner, which is formed by patterning one-dimensional (1D) ribbon array directly onto the semiconducting HPhP vdW α -MoO₃ flake with a centimeter lateral size while maintaining thicknesses of 100 ~ 200 nm. The THz and LWIR photons from far-field illuminating onto the tuner will strongly couple with the phonons and give rise to polaritons with in-plane hyperbolicities. This makes an in-plane tuner an actual device that acts not only with functions of grating but also as a polarizer and notch filter in the LWIR and THz regimes, and have distinctive features including high-Q (300) resonance and extinction ratios up to 6.5 dB at a deep sub-wavelength thickness of 200 nm. Moreover, the polariton resonance frequency, *i.e.*, the operation frequency of the polarizers and notch filters can be highly tuned by varying the period and the skew angle of the ribbon array."

C2: It is also well known that one can induce polaritonic resonances with far-field excitation by

patterning a polaritonic material into an array of nanoresonators [Nat Photonics 7, 394 (2013), Nano Lett 13, 2541 (2013)].

RC2: We totally agree with the reviewer that patterning a polaritonic material into an array of nanoresonators is a common way to induce polaritonic resonances with far-field excitation. Indeed, such far-field excitation and corresponding spectroscopic characterization, along with near-field optical nano-imaging technique, are two of the most widely used methods for fundamental characterizations of polaritonic materials. Several previous studies have reported the far-field excitation and characterization of polaritons by structuring the vdW crystal into ordered micro- or nanostructure arrays. These studies focused on plasmon polaritons with in-plane isotropic dispersion (graphene, Nat. Nanotechnol. 2011, 6, 630.; Nat. Photonics 2013, 7, 394.; Nano Lett. 2013, 13, 2541.) and hyperbolic dispersion (WTe₂, Nat. Commun. 2020, 11, 1158.), as well as PhPs with in-plane isotropic dispersion (hexagonal boron nitride, Nat. Commun. 2014, 5, 5221.). However, **far-field excitation and characterization of the PhPs with in-plane hyperbolic dispersions in vdW crystals, such as α -MoO₃ and α -V₂O₅, remain unexplored.**

In comparison with other types of polaritons, the in-plane HPhPs have the advantages of a much longer lifetime, anisotropic electromagnetic coupling, and nanoscale guiding of electromagnetic waves (Adv. Mater. 2018, 30, 1705318.; Nature 2018, 562, 557.; Sci. Adv. 2019, 5, eaav8690.). These will greatly benefit the photonic applications, especially those involving polarization control (polarizers, filters, *etc.*). Despite these beautiful visions, to the best of our knowledge, so far the demonstration of photonic applications of the in-plane HPhPs is not given. Although negative refraction and in-plane focusing of electromagnetic waves have been recently demonstrated (Science 2023, 379, 558.; Adv. Mater. 2022, 34, 2104164.), these studies all rely on exciting the polaritons with the metallic nanotip in a s-SNOM system, which is meaningful for fundamental research but not favorable for device applications. For most of practical device applications, direct excitation of the HPhPs from the far-field is necessary. Structuring the vdW crystal into ordered micro- or nanostructure arrays is a standard method for far-field excitation of the HPhPs. Nevertheless, **currently the α -MoO₃ flakes, either obtained from mechanical exfoliation or bottom-up growing, cannot achieve a large lateral size while maintaining a nanoscale thickness.** Therefore, it is a big challenge for realizing array structures that are suitable for far-field polaritonic spectroscopy on the α -MoO₃ flakes.

In our current study, we tackle this challenge by specially developing a thermal physical vapor deposition method for growing vdW α -MoO₃ with a lateral size of centimeter scale and a thickness of 100 ~ 200 nm. Accordingly, we were able to pattern 1D ribbon arrays onto the large α -MoO₃ flake and **demonstrate for the first time far-field excitation and far-field characterization of HPhPs with in-plane hyperbolicity.** Moreover, by taking advantage of the broadband light source integrated in the FTIR spectrometer used in our study (Vertex 70v, Bruker), we were able to probe not only the HPhPs in the LWIR regime (545 – 1010 cm⁻¹), but also those in the THz regime (230 – 400 cm⁻¹). The in-plane IFCs of the HPhPs in these bands

were all revealed experimentally, which can greatly help further our understanding of the HPhPs in the α -MoO₃ compared to the previous studies based on near-field measurements. Most importantly, in addition to reveal the far-field spectroscopic characteristics of the polaritonic response of α -MoO₃, we further demonstrated the practical application of the HPhPs in α -MoO₃ as a set of PNFs operating in the LWIR and THz spectral regimes. We showed that due to the salient high-Q and polarization-sensitive HPhPs resonances in the 1D ribbon arrays, tunable PNFs with a maximum peak extinction of 6.5 dB and an FWHM as narrow as 17 cm⁻¹ can be obtained.

In summary, we believe that in our current study we demonstrated for the first time far-field excitation and far-field characterization of the in-plane hyperbolic phonon polaritons in α -MoO₃ 1D nanoribbon arrays, and also, for the first time demonstrated the possible device applications of the in-plane hyperbolic phonon polaritons as polarizing notch filters in far-field optics. Our method can be extended to the far-field excitation of phonon polaritons in other vdW materials with in-plane hyperbolic dispersion.

To clarify the challenge of excitation of the HPhPs in α -MoO₃ from the far-field, a discussion has been added in paragraph 3 in page 2 as,

"...Far-field excitation and characterization of the tunable in-plane HPhPs in vdW crystals, especially in THz spectral regime, remain unexplored. Furthermore, exploring the applications of the in-plane HPhPs in optical devices has so far remained elusive. These can be done if one has a vdW crystal with large enough lateral size while maintaining thicknesses of nanometer scale, so that patterns are made larger than the diffraction limit for these free-space wavelengths and thus, suitable for far-field spectroscopy."

In addition, a discussion strengthening the advance of our study in comparison with a previous result (Nat. Commun. 2020, 11, 1158.) focusing on the far-field characterizations of vdW WTe₂ flakes with tunable in-plane hyperbolic plasmon polaritons has been added to the first paragraph in Conclusions and prospects part as,

"...It is noted that in comparison with the recently reported patterned vdW WTe₂ flakes with tunable in-plane hyperbolic plasmon polaritons at cryogenic conditions⁴⁵, the polariton tuners proposed in the current study will be more advantageous for practical applications because of their room-temperature operation, broader spectral range, and much higher quality factors of the resonance modes."

C3: Nanopatterning of van der Waals crystals and measuring the collective response of the numerous resonators in the far-field introduce undesirable artifacts such as rough edges and inhomogeneous broadening. Therefore, for fundamental characterization, a near-field study on an unpatterned crystal is preferred over the far-field measurement on an array of many resonators.

RC3: We agree with the reviewer that the near-field microscopy integrated with a broadband infrared light source, *i.e.*, the infrared nanoscopy technique, has been indeed widely used in the studies of polaritonic materials such as graphene (Nano Lett. 2011, 11, 4701.), hBN (Science 2014, 343, 1125.), α -MoO₃ (Nature 2018, 562, 557.), and α -V₂O₅ (Nat. Mater. 2020, 19, 964.). The polaritonic dispersion relation and lifetime can be derived. However, the output of the infrared source in most commercially available s-SNOM system is restricted at the spectral range over 670 cm⁻¹ due to the limitation of current manufacturing technology, while the broadband polaritonic responses of α -MoO₃ resides in a spectral range down to 260 cm⁻¹. Although s-SNOM integrated with synchrotron infrared source is able to probe the polaritons in an ultrabroadband range from 440 – 1020 cm⁻¹ (ACS Nano 2022, 16, 3027.), it is not accessible to ordinary research group. Therefore, the commercially available broadband infrared source for infrared nanoscopy cannot cover the full Reststrahlen bands of α -MoO₃, making the characterization of PhPs in these bands difficult and challenge. Furthermore, the relatively weak power of the broadband infrared source at the edge frequency (*i.e.*, ~ 670 cm⁻¹) makes the signal-to-noise ratio of the spectrum weak, thereby affecting the accuracy and authenticity of the data. Particularly, a high-power broadband THz source remain challenge, making the near-field spectroscopic characterizations of THz polariton properties difficult. In contrast, in our far-field measurements, the light source integrated in the Bruker FTIR spectrometer (Vertex 70v) operates in a broad spectral range covering the LWIR and THz regimes, which guarantees the characterizations of all the HPhPs in the patterned α -MoO₃ flake. Another issue is the measurement time consumption. For infrared nanoscopy technology, it usually takes a long time to locate, focus, and measure the sample under the microscope (an atomic force microscope). Usually, it will take half an hour or more to collect the broadband spectrum of a typical sample. Contrarily, for far-field FTIR characterization, it only takes a few minutes for the same spectral measurement.

In these regards, due to its broadband spectral range, facile operation, easy sample preparation, and high efficiency, the far-field spectroscopic characterizations of the HPhPs using the α -MoO₃ 1D ribbon arrays can greatly facilitate the study of the polaritonic properties. **More importantly**, in addition to reveal the far-field spectroscopic characteristics of the polaritonic response of α -MoO₃, another main purpose of our study is to demonstrate the possible practical application of the HPhPs in α -MoO₃. Direct far-field excitation and characterization of the HPhPs can on hand directly obtain parameters related to the photonic applications, such as polarization characteristics, optical filtering characteristics, Q-factors, *etc.*, and on the other hand directly demonstrate the device application, just as the PNFs shown by our study.

To highlight the advantages of the far-field spectroscopy characterizations with the ribbon arrays, a discussion has been added in paragraph 2 in page 3 in the revised manuscript as,

"...It should be noted that usually three main techniques are employed for determining the broadband polaritonic properties of 2D crystals, including the FTIR^{36,47,48}, electron energy loss

spectroscopy (EELS)⁴⁹, and infrared nanoscopy^{4,9,28}. In comparison with the latter two techniques which usually require a complex instrumentation, harsh sample preparation, and time consumption, the far-field polariton characteristics of the 1D-PRPs can be readily measured in a common broadband FTIR spectrometer at ambient conditions, with low time-consuming, a high collection efficiency, and over a large sample area."

We also agree with the reviewer that the fabrication of ribbons can lead to damage of the ribbon edges, which can introduce inhomogeneous broadening and lowering the Q factors of the 1D ribbon arrays. This issue is inevitable during patterning of the vdW crystal for various characterizations and device applications. To evaluate the polariton damping in the α -MoO₃ 1D ribbon arrays, we extract the lifetime of HPhPs in the array structures by fitting the reflection spectra with a Fano line-shape function (Fig. R1a),

$$F(\omega) = \frac{2p}{\pi\Gamma(q_f^2 + 1)} \frac{\left[q_f + \frac{2(\omega - \omega_0)}{\Gamma} \right]^2}{\left[1 + \frac{4(\omega - \omega_0)^2}{\Gamma^2} \right]} \quad (\text{R1})$$

where Γ is the linewidth of a specific Fano resonance, p is the amplitude, q_f is the Fano parameter accounting for the line-shape, and ω_0 is the resonance frequency. The lifetime τ of the HPhPs at resonance frequency ω_0 can be derived as $\tau = 2\hbar/\Gamma$. As shown in Fig. R1c, the polariton lifetime of the 1D ribbon arrays ranges from 0.2 ps to 3.0 ps for Reststrahlen Band 1 to Band 3.

As a comparison, we also performed near-field nano-imaging measurements on the same α -MoO₃ flake in the unpatterned region. It should be noted that due to the limitation of the quantum cascade lasers integrated into our s-SNOM system, the near-field measurements can only be conducted in the frequency range of 890 to 1250 cm⁻¹ (Band 2 and 3). In addition, to make a direct comparison, the excitation frequency of the s-SNOM was set close to the resonance frequency of the 1D ribbon arrays measured in the far-field. Specifically, for an arrays with the ribbon longitudinal axis pointing along [100] ([001]) direction, the resonance frequency was measured from far-field as ω_0 . The same frequency ω_0 was then employed as the excitation frequency for the near-field measurements. Consequently, clear interference fringes can be observed along the [001] ([100]) crystallographic direction (Fig. R1b). According to the line profiles extracted from the near-field interference fringes (Fig. R1b, insets), it is able to obtain the propagation length of the HPhPs upon an excitation frequency of ω_0 . Specifically, a typical line profile along [100] crystallographic direction of α -MoO₃ can be fitted using the equation (Nature 2018, 562, 557.),

$$y = Ax^{-0.5} e^{-\frac{x}{t_0}} \sin\left(\pi \frac{x - x_c}{w}\right), \quad A > 0, \quad w > 0, \quad t_0 > 0. \quad (\text{R2})$$

where t_0 denotes the polariton propagation length. The group velocity v_g of the HPhPs in α -MoO₃, which is defined as $v_g = \partial\omega/\partial k$, can be extracted from the polariton dispersion relation

calculated using the analytical model (Supplementary Note S6 of the revised manuscript). Afterwards, the lifetime of the HPhPs in the unpatterned α -MoO₃ can be calculated as $\tau = L/v_g$.

As shown in Fig. R1c, in comparison with the unpatterned α -MoO₃, the phonon polariton lifetime in the ribbon arrays are reduced by 20%–57% (please see also Table R1 for more details). Such a lifetime reduction is attributed to additional scattering by the rough edges of the ribbons or impurities introduced during the fabrication process. Despite the lifetime reduction, the 1D ribbon arrays still exhibit high Q factors upto 300. The lifetime of HPhPs in the α -MoO₃ ribbon arrays can be further improved by optimizing the manufacturing parameters.

Figure R1. (a) Fano lineshape fitting on a typical reflectance spectrum of the α -MoO₃ 1D ribbon arrays. The fitting was performed around the reflectance peaks and valleys. The experimental data points (symbols) are overlaid onto the fitting curve. (b) Experimental near-field images of unpatterned α -MoO₃ at the resonance frequencies of $\omega_0 = 923 \text{ cm}^{-1}$ (upper panel) and 993 cm^{-1} (lower panel). Scale bars: $0.5 \mu\text{m}$. Insets: near-field line profiles along the [100] crystallographic direction (solid red lines) of α -MoO₃. (c) Comparison of the polariton lifetimes of the unpatterned α -MoO₃ (opened symbols) and ribbon arrays (solid symbols) at various resonance frequencies. The thickness of the α -MoO₃ is $d = 200 \text{ nm}$.

Table R1. Comparison of the polariton lifetimes measured from the unpatterned α -MoO₃ and 1D ribbon arrays.

Frequencies (cm^{-1})	A: lifetime from unpatterned α -MoO ₃ (ps)	B: lifetime from 1D ribbon arrays (ps)	Difference: (A-B)/A
	Line Profile of near-field images along [001]	Ribbon longitudinal axis // [100]	
636	--	0.37	--
652.5	--	0.37	--
664	--	0.36	--
688	--	0.31	--
714.8	--	0.21	--
745	--	0.20	--
993	3.62	2.40	34%
996	3.71	2.63	29%
998	3.99	3.00	23%

999	3.50	2.80	20%
1000	3.40	2.60	23%
Frequencies (cm ⁻¹)	C: lifetime from unpatterned α -MoO ₃ (ps)	D: lifetime from 1D ribbon arrays (ps)	Difference: (C-D)/C
	Line Profile of near-field images along [100]	Ribbon longitudinal axis // [001]	
906	0.89	0.55	38%
912	1.15	0.59	48%
915	1.31	0.61	53%
917	1.71	0.73	57%
923	1.23	0.93	25%
993	4.03	2.62	35%
996	3.96	2.46	38%
998	4.32	2.65	38%
999	3.91	2.55	34%
1000	3.83	2.41	37%

To clarify the additional damping induced by fabrication of the ribbons, a discussion is added in paragraph 3 in page 8 as,

"The far-field spectra also allow for extracting the polariton lifetimes, which span from 0.2 to 3.0 ps in the three Reststrahlen bands (Supplementary Note S5, Fig. S9, and Table S1). It is noted that fabrication of 1D-PRPs can lead to damage of the ribbon edges, which can reduce the polariton lifetimes and Q factors. Usually this issue is inevitable during patterning of the vdW crystal for various characterizations and device applications. To evaluate the additional polariton damping induced by the patterning processes, we performed near-field measurements on the same α -MoO₃ flake in the unpatterned region and extracted the intrinsic polariton lifetimes (Supplementary Note S5, Fig. S9, and Table S1). In comparison with the unpatterned α -MoO₃, the phonon polariton lifetime in the ribbon arrays are reduced by 20%–57%. Such lifetime reduction is attributed to additional scattering by the rough edges of the ribbons or impurities introduced during the fabrication process. Despite the lifetime reduction, the 1D-PRPs still exhibit high Q factors upto 300. Although such a value is smaller than that of resonances sustained by an unpatterned and naturally grown α -MoO₃ ribbon⁶¹, it can be further improved by optimizing the processing parameters."

In addition, the detailed discussion on extraction of the polariton lifetimes from far-field and near-field measurements has been added as Note S5 in Supplementary Information part as,

"Supplementary Note S5. Extraction and comparison of the polariton lifetimes obtained from far-field and near-field measurements

We extracted the lifetime of HPhPs in the 1D-PRPs by fitting the reflection spectra with the Fano line-shape function (Eq. S1 and Fig. S9a). The lifetime τ at a resonance frequency ω_0 can be derived as $\tau = 2\hbar/\Gamma$. As shown in Fig. S9c, the polariton lifetime of the 1D-PRPs ranges from 0.2 ps to 3.0 ps for Reststrahlen Band 1 to Band 3. We also performed near-field nano-imaging

measurements on the same α -MoO₃ flake in the unpatterned region. A scattering type scanning near-field optical microscope was employed for the near-field characterizations (neaSNOM, neaspec GmbH)^{6,7}. It should be noted that due to the limitation of the quantum cascade lasers integrated into our s-SNOM system, the near-field measurements can only be conducted in the frequency range of 890 to 1250 cm⁻¹ (Band 2 and 3). In addition, to make a direct comparison, the excitation frequency of the s-SNOM was set close to the resonance frequency of the 1D-PRPs measured in the far-field. Specifically, for an 1D-PRP with the ribbon longitudinal axis pointing along [100] ([001]) direction, the resonance frequency was measured from far-field as ω_0 . The same frequency ω_0 was then employed as the excitation frequency for the near-field measurements. Consequently, clear interference fringes can be observed along the [001] ([100]) crystallographic direction (Fig. S9b). According to the line profiles extracted from the near-field interference fringes (Fig. S9b, insets), it is able to obtain the propagation length of the HPhPs upon an excitation frequency of ω_0 . A typical line profile along [100] crystallographic direction of α -MoO₃ can be fitted using the equation⁸,

$$y = Ax^{-0.5} e^{-\frac{x}{t_0}} \sin\left(\pi \frac{x-x_c}{w}\right), \quad A > 0, \quad w > 0, \quad t_0 > 0. \quad (\text{S13})$$

where t_0 denotes the polariton propagation length. The group velocity v_g of the HPhPs in α -MoO₃, which is defined as $v_g = \partial\omega/\partial k$, can be extracted from the polariton dispersion relation calculated using the analytical model (see Note S6 below). Afterwards, the lifetime of the HPhPs in the unpatterned α -MoO₃ can be calculated as $\tau = L/v_g$.

The comparison of the polariton lifetimes obtained from far-field and near-field measurements is shown in Fig. S5c and Table S1."

Fig. R1 is added as Fig. S9 in the Supplementary Information part. Table R1 is added as Table S1 in the Supplementary Information part. The reference "ACS Nano 2022, 16, 3027." is added as Ref. 61 in the references list.

C4: Polaritonic resonators based on α -MoO₃ have been recently studied and the reported Q factors are above 700, which is more than two times the value reported in this paper [ACS Nano 16, 3027 (2022)].

RC4: The main purpose of our current study is to study the far-field characteristics of HPhPs in α -MoO₃ and to demonstrate the possible photonic applications. All of the polariton resonances were characterized using the far-field spectroscopy. Therefore, only those resonance modes with high out-coupling efficiencies to the far-field, *i.e.*, high radiation rates, will be probed and analyzed. In other words, in our current study we only focused on the HPhPs resonances with large radiation losses. In contrast, in the reference mentioned by the reviewer (ACS Nano 2022, 16, 3027.), the polariton resonances were probed by s-SNOM technique. A metallic nanotip was employed to focus the incident light for excitation of the polariton resonance modes and then out-couple the polariton radiation for measurement. As mentioned by the authors, the Q-factors of the polariton resonances are strongly dependent on the position where the near-field scattering

spectra were taken. Specifically, they showed that the polariton resonances measured near the α -MoO₃ ribbon edge exhibited a high Q-factors upto 718, which is much higher than those measured at the center of the ribbon. They owed the high-Q resonance at the edges to higher order Fabry–Pérot (FP) mode, while the low-Q resonance at the center to the lowest order FP mode. The difference in the Q-factors can be ascribed to the variation of the radiation losses of the different FP modes along the ribbon axis. In comparison with the high-order modes at the edge, the lowest-order mode has relatively strong E_z components at the ribbon center that facilitate enhanced radiative out-coupling losses *via* the probe tip. Consequently, the resonance modes exhibited a lower Q-factors of 444 at the ribbon center. In this regard, we argue that the polariton resonances measured from the far-field are mainly contributed by the lowest-order FP modes at the ribbon center measured from the near-field.

The highest Q-factor measured in our study (300 for resonance modes in Band 3) is about 48% smaller than that obtained from the ribbon center reported in "ACS Nano 2022, 16, 3027." (444 for resonance modes in Band 3 as well). This is because the Q-factor reported in the previous study is an intrinsic value for a naturally grown nanoribbon placed onto an ultrasmooth gold substrate. The crystallinity of the ribbon is excellent and the edges are smooth, whereby the polariton damping can be reduced to a minimum. In contrast, in our study, the ribbon arrays were fabricated by patterning a large α -MoO₃ flake. The fabrication processes will inevitably damage the ribbon edges and introduce additional damping for the polariton resonances. In addition, the ribbon arrays in our study were placed onto silicon substrate with a 300-nm oxide layer. In comparison with the gold substrate, the SiO₂ layer can introduce additional damping pathways for the polaritons in α -MoO₃ (Adv. Optical Mater. 2022, 10, 2201492.). These issues will lower the Q factors of the resonances measured from far-field. According to our analyses (please refer to the response RC3 above), in comparison with the unpatterned α -MoO₃, the phonon polariton lifetime in the ribbon arrays are reduced by 20%–57%. The reduction of Q-factors in the 1D ribbon arrays (48%) is within this range. We believe that the Q-factors of the ribbon arrays can be improved by optimizing the processing parameters.

We would like to emphasize again that one of the main purposes in our current study is to demonstrate the possible practical application of the HPhPs in α -MoO₃. In comparison to the single resonator reported in "ACS Nano 2022, 16, 3027.", the ribbon array reported in our study allows us to modulate the wavefront of the incident light and control their power flow in an engineered space, which therefore enables a variety of interesting applications, such as negative refraction, holography, metalens, polarization conversion, and even topological PhPs and exciton polaritons with robust beam steering functionalities, in visible and near-infrared ranges to be expanded into the THz and LWIR spectral regimes. Therefore, despite the relatively low Q-factor compared with the previous reported values, we strongly believe that our results advance toward the development of nanophotonic devices using HPhPs vdW materials.

To clarify the Q-factor of the 1D ribbon array, the discussion in paragraph 2 in page 8 has been modified as,

"...In particular, the Q-factor of the Band 3 resonance can be as high as 300, which is on par with the highest observed in hBN nanoresonators (360) via the same far-field technique⁶⁰...."

Response to Reviewer #2

We thank this reviewer for the positive comments on our manuscript and the insightful suggestions.

***Comment:** Wuchao Huang and co-authors reported a far-field excitation of in-plane hyperbolic polaritons in α -MoO₃ crystals by structuring the medium with momentum-matched periodic grating patterns. The authors demonstrated that the device works as an in-plane hyperbolic phonon-polariton tuner and exhibits a unique set of characteristic functions such as polarizer and notch filter for LWIR and THz regimes. This is the first demonstration of far-field control of high-k in-plane hyperbolic phonon polaritons in the LWIR and THz regimes. I believe this study will drive the field of photonics and the advancement of 2D materials for optoelectronics in LWIR and the THz region. The analytical and numerical simulations well supported the experimental findings. The results are convincing, timely, and essential to the field of mid-infrared photonics and 2D materials. I believe the manuscript is worthy of publication in Nature Communications, provided a suitable (but minor) revision is made to support the authors' claims. The list of my questions/suggestions is provided below.*

***Q1:** I believe the authors' reported study of in-plane hyperbolic polariton tuner is conceptually very similar to the earlier theoretical and experimental demonstrations of "Hyper Crystals" derived from structured plasmonics hyperbolic metamaterials (Narimanov, Evgenii E. "Photonic hypercrystals." Physical Review X 4.4 (2014): 041014., Galfsky, Tal, et al. "Photonic hypercrystals for control of light-matter interactions." Proceedings of the National Academy of Sciences 114.20 (2017): 5125-5129.). Authors may cite these first demonstration papers.*

RI: According to the reviewer's suggestion, in the section "Fabrication of in-plane hyperbolic polariton tuner and far-field excitation of HPhPs", we cited the two references when first mentioning the far-field excitation of the HPhPs using the one-dimensional periodic ribbon patterns (1D-PRPs) (paragraph 2 in page 4). The statement now becomes,

"A tuner comprised of 1D-PRPs (Fig. 1c and Supplementary Fig. S2) is able to overcome the large momentum mismatch^{52,53}...."

The two references mentioned by the reviewer are added as Ref. 52 and 53 in the reference list.

Q2: The authors patterned the α -MoO₃ with periodic grating patterns for far-field excitation of high momentum in-plane hyperbolic phonon-polaritons. These hyperbolic phonon-polaritons can be both surface states and bulk states. The near-field technique (NSOM) only excites and images the surface phonon-polaritons. However, in the current study, the outcoupled modes' character is not well-defined whether they are surface or bulk modes.

R2: As mentioned by the reviewer, it is indeed that there can exist both of surface and volume phonon polaritons (PhPs) in the α -MoO₃ flake. The observed PhPs in the van der Waals (vdW) hyperbolic thin flakes (with thickness of 100 ~ 200 nm) are actually bulk modes, whose electromagnetic fields are confined within the body of the flakes (please refer to our previous studies and that from another group, Adv. Mater. 2018, 30, 1705318.; Sci. Adv. 2019, 5, eaav8690.; Nature 2018, 562, 557.). Due to the finite thickness of the α -MoO₃ flakes, these bulk modes can be launched and detected by the s-SNOM technique. The origin of the resonances observed in the far-field reflectance spectra of the 1D-PRPs can be unveiled by calculating their corresponding near-field electromagnetic distributions. For example, for a 1D-PRP with a ribbon width $w = 800$ nm and orientated along the [001] axis of α -MoO₃, two resonances can be observed at $\omega = 874$ and 1001 cm⁻¹ upon excitation polarized along the [100] axis, which are respectively manifested as a reflectance peak and a valley (Fig. 1e in our original manuscript). As shown in Fig. R2, in the cross section perpendicular to the long axis of the ribbon, the electric field distributions at these two frequencies clearly reveal "zig-zag"-shape polaritonic rays propagating inside the ribbon. These are typical characteristics of HPhPs bulk modes, which have been previously revealed in hBN nanostructures (Nat. Commun. 2014, 5, 5221.) and α -MoO₃ flakes using s-SNOM technique (Adv. Mater. 2018, 30, 1705318.; Sci. Adv. 2019, 5, eaav8690.; Nature 2018, 562, 557.).

Figure R2. Simulated optical near-field distributions $|E(x, z)|$ of a typical α -MoO₃ 1D-PRP at 874 (a) and 1001 cm⁻¹ (b). The near-field distributions are drawn on the cross section perpendicular to the ribbon long axis, *i.e.*, the x - z plane. Scale bars: 400 nm. The grey dashed arrows indicate the propagation trajectories of the polariton rays.

A discussion clarifying the polariton modes observed in the far-field spectra were added in paragraph 3 in page 4 as,

"...These are typical fingerprints of HPhP waveguide modes. These modes are bulk modes with electromagnetic fields confined inside the body of the flakes, such as those observed in hBN nanostructures⁵⁵ and biaxial α -MoO₃ flakes⁵⁶...."

Q3: The experiments were performed using a commercially available Schwarzschild objective in the FTIR equipment. The NA of this objective is usually 0.58, implying that measured reflections correspond to averaged responses over +/- 35°. At the same time, the reflection response from the analytical simulations was performed for each angle. I believe there should be a discrepancy in the theory and experimental results matching. However, the authors claim a good agreement between the theory and experiments. Can the authors explain the reason for no mismatch?

R3: We agree with the reviewer that, in principle, the experimental reflection or transmission spectra collected in FTIR measurement is the overall contribution of the reflected/transmitted light over a finite solid angle range, which is defined by the N.A. of the objective. In our measurement, a 15× Schwarzschild objective of N.A. ~ 0.4 was used to focus the light onto the sample and collect the reflection light. In this case, the measured spectra in our data corresponds to the averaged responses over +/- 24°.

When referring a good agreement between the theory and experiments, we actually intend to claim that a good agreement is obtained between the polariton dispersion (dispersion relation and the isofrequency contour: IFC) derived from experimental measurements and from the analytical simulation, but not between the experimental reflection spectra and theoretical reflection spectra. However, it should be noted that the experimental reflection spectra indicate the reflection response of a **patterned α -MoO₃ one-dimensional (1D) ribbon array**, while the analytical calculated Fresnel coefficient correspond to the reflection response of a **flat and homogeneous flake with a finite thickness**.

To obtain the polariton dispersion using the analytical model, reflection spectra under p -polarized illumination at normal incidence, $r_p(q_{\text{PhPs}}, \omega)$, for each possible polariton momentum and frequency are first calculated. Because the imaginary part of $r_p(q_{\text{PhPs}}, \omega)$, $\text{Im}r_p(q_{\text{PhPs}}, \omega)$, represents the absorption efficiency and therefore, the excitation efficiency of HPhPs (Nano Lett. 2011, 11, 4701.; Science 2014, 343, 1125.). The polariton dispersion relations and IFCs can then be obtained according to the maxima of $\text{Im}r_p(q_{\text{PhPs}}, \omega)$, as shown by the brightest trajectories illustrated in Fig. 2c and Fig. 3c–f in our original manuscript.

In the experimental measurements, as we discussed in the manuscript, due to the large photon–polariton momentum mismatch, the HPhPs cannot be directly excited from far-field (even at oblique illumination with large incident angle). In our study, such a mismatch can be compensated by structuring the α -MoO₃ flake into 1D ribbon arrays, where the periodic corrugations can scatter the incident plane wave into evanescent waves carrying a momentum of

$q = k_0 \sin \alpha \mp G \approx \mp G = \mp \frac{\pi}{w}$, with w the ribbon width, α the incident angle, and k_0 the free-space

photon momentum. In this way the HPhPs can be efficiently excited, giving rise to a resonance peak in the reflection spectrum. The resonance frequency is strongly dependent on w , and therefore, by extracting the resonance frequencies ω_0 from the various ribbon arrays with different w , it is able to derive the polariton dispersion relations. The IFCs can also be drawn by extracting the resonance frequencies ω_0 from the various ribbon arrays with different skew angle θ while fixed w . We compared these experiment polariton dispersions with the analytical ones, and found that the experimental data points overlaid onto the brightest trajectories of $\text{Im}r_p(q_{\text{PhPs}}, \omega)$, we then said a good agreement between the theory and experiments. It is noted that even though the ribbon arrays were illuminated at an incident angle of $\alpha = 24^\circ$, because

$k_0 \sin \alpha \ll \mp \frac{\pi}{w}$, the phase matching condition $q = k_0 \sin \alpha \mp G \approx \mp G = \mp \frac{\pi}{w}$ will not be changed.

Thus, as long as the w and θ are fixed, the resonance frequencies ω_0 observed in the far-field reflectance spectrum will be the same for different illumination and collection angles. This is the reason for the good agreement between the theory and experiment.

Q4: The authors mentioned they determined the bulk phonon modes of MoO₃ from Raman scattering measurements. Do these in-plane hyperbolic phonon-polariton modes also show the Raman scattering signal? I believe the authors should report these measurements to claim the phononic nature of these modes.

R4: The reviewer raised a very interesting issue. A phonon mode is the vibration mode of the crystal lattice, while a PhPs mode is a hybridized mode induced by coupling of the incident photon with the intrinsic phonon. The transverse optical phonon mode with Raman activity (determined by the Raman scattering tensor) in a specific crystal can be determined by Raman scattering measurements. In our previous study (J. Phys. Chem. C 2021, 125, 765.), the in-plane anisotropic phonon modes in the vdW α -MoO₃ flake has been systematically studied, which exhibit strong polarization sensitive properties (please also refer to Fig. S1 in Supplementary Information of the revised manuscript).

For the PhPs, they are sustained within a broad frequency bands bracketed by the transverse and longitude optical phonon frequencies. The momenta of the PhPs are basically much larger than the excitation photon. As a result, usually the PhPs modes cannot be directly detected from the traditional Raman spectroscopy, which employs a far-field excitation manner. This can be done by introducing a specific measurement configuration. For example, it has been reported that the PhPs in GaP can be probed by Raman spectroscopy with oblique incident configuration (Phys. Rev. Lett. 1965, 15, 964.). This configuration is limited to measure polaritons with relatively small momenta. For high-momentum PhPs, it is a challenge for characterizing the Raman scattering from the PhPs, and one needs to employ scanning probe microscope integrated with a

metallic nanotip (*e.g.*, tip-enhanced Raman spectroscopy) for exciting the polaritons and measuring the scattering spectrum. Such a technique is currently unavailable in our scanning near-field optical microscope (SNOM).

The main purposes of our current study is: 1) exciting and characterizing the in-plane hyperbolic PhPs from the far-field by using 1D α -MoO₃ ribbon arrays.; 2) demonstrating the possible applications of the ribbon arrays as polarizing notch filters in the long-wave infrared (LWIR) and terahertz (THz) spectral regimes. We are trying our best to improve and upgrade our SNOM system and study the Raman scattering of the PhP modes in our future work.

Q5: In Fig. 1c, the authors show the SEM images of the patterns drawn at different skew angles. The pattern image for 30° skew angle seems to correspond to 60° and vice versa. I believe it is only a mistake, and actual measurements reported in Fig. 3 were not performed following this notation.

R5: We apologize for this mistake. In the revised manuscript, the order of skew angles in Fig. 1c has been revised, as shown in Fig. R3.

Figure R3. Far-field excitations of HPhPs in in-plane hyperbolic polariton tuners. (a) Schemes of the tuners comprised of vdW α -MoO₃ periodic ribbon patterns and FTIR measurements. (b) Photograph of a typical α -MoO₃ flake grown on silicon substrate with a

300-nm oxide layer. The largest lateral length of the flake is 1 cm. (c) Scanning electron microscopy image of periodic ribbon patterns with a fixed $w = 800$ nm and different skew angles θ of 0° , 30° , 60° , and 90° . (d) Polarized reflectance spectra of the pristine α -MoO₃ thin flake shown in (b). The electric fields of the incident light, E_{inc} , are along [001] (red curve) and [100] (blue curve) crystallographic directions, respectively. (e, f) Experimental polarized reflectance spectra of one-dimensional periodic tuner patterns with $\theta = 0^\circ$ (e) and 90° (f). The polarization of the incident light is paralleled to [001] (red curves) and [100] (blue curves) directions, respectively. Insets: enlarged reflectance spectra in the range of 995 to 1020 cm^{-1} .

Q6: I suggest the authors cite the recent developments of phonon and exciton-polaritons in 2D materials, especially the topological phonon-polaritons (Guddala, S. et al. "Topological phonon-polariton funneling in midinfrared metasurfaces." Science 374.6564 (2021): 225-227.), and topological exciton-polaritons (Li, Mengyao, et al. "Experimental observation of topological Z2 exciton-polaritons in transition metal dichalcogenide monolayers." Nature communications 12.1 (2021): 1-10.).

R6: As suggested by the reviewer, the two references "Science 2021, 374, 225." and "Nat. Commun. 2021, 12, 4425." have been cited as Ref. 68 and 69 in the revised manuscript. The corresponding discussion has also been added in paragraph 4 in page 15 as,

"...The tuner allows us to modulate the wavefront of the incident light and control their power flow in an engineered space²¹⁻²⁶, which therefore enables a variety of interesting applications, such as negative refraction, holography, metalens, polarization conversion, and even topological PhPs and exciton polaritons with robust beam steering functionalities^{68,69}, in visible and near-infrared ranges to be expanded into the THz and LWIR spectral regimes."

Q7: I suggest the authors increase the text's font in all figures.

R7: As suggested by the reviewer, the text fonts in all figures, including those in the main text and Supplementary Information, are increased.

Q8: The authors should review Fig. S3. The THz region Reststrahlen bands were labeled reverse to the LWIR region bands.

R8: We apologize for the mistakes. The Supplementary Fig. S3 has been revised as Fig. R4 in

the revised manuscript.

Figure R4. Real parts of permittivity of vdW α - MoO_3 along the three crystallographic axes.

Response to Reviewer #3

We thank this reviewer for the rigorous comments on our manuscript and the insightful suggestions.

***Comment:** The authors report far-field (far-)infrared spectroscopy of ribbon arrays fabricated from large MoO_3 flakes. Results seem interesting and in some aspect new, but reading the manuscript the reviewer was sometimes confused by the description of the work. The authors do more or less the same as many others were doing before: fabricating ribbons from van der Waals materials/2D material/graphene/etc and then performing far-field infrared and THz spectroscopy. But the authors stress a certain challenge in doing so (although it is quite standard), and much more confusing, they introduce the expression “polariton tuner” which the reviewer does not understand well. The authors do very similar experiments as reported e.g. in ref. 50, just probing phonon polaritons instead of plasmon polaritons. There are also several scientific/technical details that need clarification. Details see below.*

In summary, although the work principally is quite interesting, I find the wording and description of the work sometimes misleading/confusing for the reader. It may become suitable for Nature Communications when the authors describe their work and results more clearly regarding technical, scientific and novelty aspects, avoiding expressions that are not well defined/explained or justified (such as lattice, tuners, etc).

We thank very much the reviewer for raising important comments on our current study. These comments can greatly help us improve the scientific quality of our study.

On the basis of our understanding, we summarize the reviewer's comments in two aspects, which we will provide point-to-point responses. Afterwards, the point-to-point responses to the scientific/technical details.

CI: The authors do more or less the same as many others were doing before: fabricating ribbons from van der Waals materials/2D material/graphene/etc and then performing far-field infrared and THz spectroscopy. But the authors stress a certain challenge in doing so (although it is quite standard), and much more confusing, they introduce the expression "polariton tuner" which the reviewer does not understand well.

RC1: We agree with the reviewer that the methodology used in our current study is similar to those in previous reports. Indeed, previous studies have reported the far-field excitation and characterization of polaritons by structuring the van der Waals (vdW) crystals into ordered micro- or nanostructure arrays. This is based on the fact that direct far-field excitation of polaritons can be achieved by patterning polaritonic materials into nanoribbon arrays, which can effectively compensate the large momentum mismatch between free-space photons and polaritons. In addition, direct far-field excitation of polaritons can greatly facilitate the practical application of corresponding polaritonic materials. We fully agree that this is a standard method for studying the far-field polaritonic responses of a specific crystal.

However, most of the previous studies focused on plasmon polaritons with in-plane isotropic (graphene, Nat. Nanotechnol. 2011, 6, 630.) and hyperbolic dispersions (WTe₂, Nat. Commun. 2020, 11, 1158.), as well as phonon polaritons (PhPs) with in-plane isotropic dispersion (hexagonal boron nitride, Nat. Commun. 2014, 5, 5221.). The far-field excitation and characterization of the PhPs with in-plane hyperbolic dispersions in vdW crystals, such as α -MoO₃ and α -V₂O₅, remain unexplored. In comparison with other types of polaritons, the in-plane hyperbolic phonon polaritons (HPhPs) have the advantages of a much longer lifetime, anisotropic electromagnetic coupling, and nanoscale guiding of electromagnetic waves (Adv. Mater. 2018, 30, 1705318.; Nature 2018, 562, 557.; Sci. Adv. 2019, 5, eaav8690.). These will greatly benefit the photonic applications, especially those involving polarization control (polarizers, filters, *etc.*). Despite these beautiful visions, to the best of our knowledge, so far the demonstration of photonic applications of the in-plane HPhPs is not given. Although negative refraction and in-plane focusing of electromagnetic waves have been recently demonstrated (Science 2023, 379, 558.; Adv. Mater. 2022, 34, 2104164.), these studies all rely on exciting the polaritons with the metallic nanotip in a s-SNOM system, which is meaningful for fundamental research but not favorable for device applications. For most of practical device applications, direct excitation and characterization of the HPhPs from the far-field is necessary. This can on one hand directly obtain parameters related to the photonic applications, such as polarization

characteristics, optical filtering characteristics, Q-factors, *etc.*, and on the other hand directly demonstrate **the possible device application, just as the polarizing notch filters (PNFs) shown by our study.**

Structuring the vdW crystal into ordered micro- or nanostructure arrays is indeed a standard method for far-field excitation of the HPhPs. A prerequisite is a vdW crystal with large enough lateral size while maintaining thicknesses of nanometer scale, so that patterns are made larger than the diffraction limit for these free-space wavelengths and thus, suitable for far-field spectroscopy. Nevertheless, **currently the α -MoO₃ flakes, either obtained from mechanical exfoliation or bottom-up growing, cannot achieve a large lateral size while maintaining a nanoscale thickness.** Therefore, it is a big challenge for realizing array structures that are suitable for far-field polaritonic spectroscopy on the α -MoO₃ flakes. In our current study, we tackle this challenge by specially developing a thermal physical vapor deposition method for growing **vdW α -MoO₃ with a lateral size of centimeter scale and a thickness of 100 ~ 200 nm.** Accordingly, we were able to pattern one-dimensional (1D) ribbon arrays onto the large α -MoO₃ flake and **demonstrate for the first time far-field excitation and far-field characterization of HPhPs with in-plane hyperbolicity.**

When introducing a "tuner", we aim to demonstrate that the 1D ribbon arrays can act as a device with functionalities of responding directly to the incident long-wave infrared and THz electromagnetic waves, and having high-quality-factor (high-Q) and tunable phonon polariton resonances with large light extinction ratios. By taking advantage of these features, the arrays can filter the polarization states as well as frequencies of the incident wave. Therefore, a specific ribbon array can act as a polarizer and filter, or more precisely as shown by our results, a polarizing notch filter (PNF) that combines a polarizer and a narrow band-rejection filter together into a single ribbon array. Moreover, we showed that the polariton resonance frequency, *i.e.*, the rejection frequency of such polarization notch filters can be highly tuned by varying the period and the skew angle of the α -MoO₃ array of ribbon. These are the functionalities of the ribbon arrays. Because the polariton resonances of the array stem from the in-plane hyperbolic phonon polaritons of the α -MoO₃, we define the ribbon array as a tunable device, and name it as "in-plane hyperbolic polariton tuner".

More discussion regarding the novelty of our current study can be found in the response to **Q1** of Reviewer #3.

C2: The authors do very similar experiments as reported e.g. in ref. 50, just probing phonon polaritons instead of plasmon polaritons. There are also several scientific/technical details that need clarification.

RC2: In Ref. 50 (Nat. Commun. 2020, 11, 1158.), plasmon polaritons with in-plane hyperbolic dispersion in vdW WTe₂ flake were reported using the far-field spectroscopy. In comparison with the in-plane hyperbolic plasmon polaritons, the in-plane hyperbolic phonon polaritons (HPhPs)

have the advantages of a much longer lifetime, anisotropic electromagnetic coupling, and nanoscale guiding of electromagnetic waves (Adv. Mater. 2018, 30, 1705318.; Nature 2018, 562, 557.; Sci. Adv. 2019, 5, eaav8690.). These will greatly benefit the photonic applications, especially those involving polarization control (polarizers, filters, *etc.*). In our current study, we managed to pattern 1D ribbon arrays onto the large α -MoO₃ flake and **demonstrate for the first time far-field excitation and far-field characterization of HPhPs with in-plane hyperbolicity**. Moreover, we further demonstrated the practical application of the HPhPs in α -MoO₃ as a set of PNFs operating in the long-wave infrared (LWIR) and terahertz (THz) spectral regimes. We showed that due to the salient high-quality-factor (high-Q) and polarization-sensitive HPhPs resonances in the 1D ribbon arrays, tunable PNFs with a maximum peak extinction of 6.5 dB and an FWHM as narrow as 17 cm⁻¹ can be obtained.

It is noted that in the patterned vdW WTe₂ flakes, the in-plane hyperbolic plasmon polaritons need to operate at cryogenic conditions to realize resonances with high-Q factors. In contrast, all of the far-field spectroscopic measurements of the 1D ribbon arrays in our study were conducted at room temperature. This is due to that the phonon polaritons were contributed by coupling of photons and lattice vibrations, where the damping from the lattice scattering will be strongly suppressed (Nature 2018, 562, 557.; Adv. Mater. 2018, 30, 1705318.; Sci. Adv. 2019, 5, eaav8690.). This room-temperature operation manner enables practical photonic applications of the vdW polaritonic crystals devices, for example, as demonstrated in our study, the polarizing notch filters (PNFs).

More discussion on the advances of our current study in comparison with the Ref. 50 is given in the response to **Q1(d)** of Reviewer #3.

Q1: *For example, the authors write the following without providing explanation or justification for the claims in the abstract:*

“Here, we propose an In-Plane Hyperbolic Polariton Tuner and demonstrate a first device responding to direct far-field excitation and giving rise to a unique set of characteristic functions at far-field.”

a) The reviewer does not see a device, just an array of ribbons.

R1a: When claiming a "device", we aim to demonstrate an architecture with some specific functionalities that are readily used in the optical path for tuning the electromagnetic wave properties. Specific to the 1D ribbon arrays reported in our current study, the architecture is the ribbon arrays. Such arrays can respond directly to the incident LWIR and THz electromagnetic waves and have PhPs resonances with high quality factors up to 300 and extinction ratios up to 6.5 dB. By taking advantage of these features, the arrays can filter the polarization states as well as frequencies of the incident wave. Therefore, a specific ribbon array can act as a polarizer and filter, or more precisely as shown by our results, a polarizing notch filter (PNF) that combines a

polarizer and a narrow band-rejection filter together into a single ribbon array. Moreover, we showed that the polariton resonance frequency, *i.e.*, the rejection frequency of such polarization notch filters can be highly tuned by varying the period and the skew angle of the 1D ribbon array. These are the functionalities of the ribbon arrays. Moreover, because the polariton resonances of the array stem from the in-plane HPhPs of the α -MoO₃, we define the ribbon array as a tunable device, and name it as "in-plane hyperbolic polariton tuner".

To clarify the device we claimed, the discussion in paragraph 4 in page 2 has been modified as,

"In this article we demonstrate for the first time far-field excitation and far-field characterization of HPhPs in an In-Plane Hyperbolic Polariton Tuner, which is formed by patterning one-dimensional (1D) ribbon array directly onto the semiconducting HPhP vdW α -MoO₃ flake with a centimeter lateral size while maintaining thicknesses of 100 ~ 200 nm. The THz and LWIR photons from far-field illuminating onto the tuner will strongly couple with the phonons and give rise to polaritons with in-plane hyperbolicities. This makes an in-plane tuner an actual device that acts not only with functions of grating but also as a polarizer and notch filter in the LWIR and THz regimes, and have distinctive features including high-Q (300) resonance and extinction ratios up to 6.5 dB at a deep sub-wavelength thickness of 200 nm. Moreover, the polariton resonance frequency, *i.e.*, the operation frequency of the polarizers and notch filters can be highly tuned by varying the period and the skew angle of the ribbon array."

b) why "first" if same has been done in e.g. ref 50 and other works?

RIb: As mentioned by the reviewer, several earlier studies have reported the far-field excitation and characterization of polaritons by structuring the vdW crystal into ordered micro- or nanostructure arrays. These previous studies focused on plasmon polaritons with in-plane isotropic (graphene, Nat. Nanotechnol. 2011, 6, 630.) and hyperbolic dispersions (WTe₂, Nat. Commun. 2020, 11, 1158.), as well as PhPs with in-plane isotropic dispersion (hexagonal boron nitride, Nat. Commun. 2014, 5, 5221.). However, far-field excitation and characterization of the PhPs with in-plane hyperbolic dispersions in vdW crystals, such as α -MoO₃ and α -V₂O₅, remain unexplored. In comparison with other types of polaritons, the in-plane HPhPs have the advantages of a much longer lifetime, anisotropic electromagnetic coupling, and nanoscale guiding of electromagnetic waves (Adv. Mater. 2018, 30, 1705318.; Nature 2018, 562, 557.; Sci. Adv. 2019, 5, eaav8690.), which will greatly benefit the photonic applications, especially those involving polarization control (polarizers, filters, *etc.*). Nonetheless, **exploring the applications of the in-plane hyperbolic phonon polaritons in optical devices has so far remained elusive.** As mentioned in our manuscript, the challenge lies in that currently the α -MoO₃ flakes are too small for patterning the array structures that are suitable for far-field spectroscopy.

According to these considerations, we believe that in our current study we demonstrated for the first time far-field excitation and far-field characterization of the in-plane HPhPs in α -MoO₃ 1D nanoribbon arrays, and also, the first time demonstration of possible device applications of the

in-plane HPhPs as polarizing notch filters in far-field optics.

As suggested by the reviewer, a discussion has been added in paragraph 3 in page 2 to clarify the novelty of our study,

"...For most of practical device applications, direct excitation of the HPhPs from the far-field is necessary. Some earlier studies indicate that it is possible to pattern the surface of vdW crystals, such as with graphene⁴⁰⁻⁴², hexagonal boron nitride^{7,43,44}, semi-metals⁴⁵, and topological insulators⁴⁶ to excite and measure the various types of polaritons. However, these previous studies focused on plasmon polaritons with in-plane isotropic^{40-42,46} and hyperbolic dispersions⁴⁵, as well as phonon polaritons with in-plane isotropic dispersion^{7,43,44}. Far-field excitation and characterization of the tunable in-plane HPhPs in vdW crystals, especially in THz spectral regime, remain unexplored. Furthermore, exploring the applications of the in-plane HPhPs in optical devices has so far remained elusive. These can be done if one has a vdW crystal with large enough lateral size while maintaining thicknesses of nanometer scale, so that patterns are made larger than the diffraction limit for these free-space wavelengths and thus, suitable for far-field spectroscopy."

c) what do the authors mean by "set of functions at far field" when just recording far-field of static ribbons?

R1c: When saying "a unique set of characteristic functions at far-field", we intended to describe that the polariton tuner, which is composed of α -MoO₃ 1D ribbon array, exhibits resonances that are of high-Q, polarization sensitive, deep extinction ratio, and tunable by modifying the ribbon width and orientation. To avoid any misleading and differentiate the proposed tuner from a device with active control (switchable to external stimuli), we modify the description of the tuner in the Abstract part as,

"...Here, we propose an In-Plane Hyperbolic Polariton Tuner that is based on patterning van der Waals semiconductors, here α -MoO₃, into ribbon arrays. We demonstrate that such tuners respond directly to far-field excitation and give rise to LWIR and THz resonances with high quality factors up to 300, which are strongly dependent on in-plane hyperbolic polariton of the patterned α -MoO₃. We further show that with this tuner, intensity regulation of reflected and transmitted electromagnetic waves, as well as their wavelength and polarization selection can be achieved...."

d) what is unique regarding the functions and why, e.g. compared to ref 50?

R1d: In comparison with Ref. 50 (Nat. Commun. 2020, 11, 1158.), which reported the far-field characterizations of hyperbolic plasmonic vdW crystal WTe₂, we believe that the α -MoO₃ 1D ribbon array, or the polariton tuner, can have four unique functions.

First, the resonance linewidth of the polariton tuner is much smaller than that of the patterned WTe_2 , making the Q-factor of the tuner an order of magnitude higher (~ 20 in patterned $\alpha\text{-MoO}_3$ vs. ~ 2 in patterned WTe_2) in the THz spectral region ($200 - 400 \text{ cm}^{-1}$). This is because the resonances in the tuner are originated from the in-plane HPhPs, while those in patterned WTe_2 is due to the hyperbolic plasmon polaritons. In comparison with the plasmon polaritons that are contributed by the interplay of interband and intraband electron transitions in WTe_2 , the PhPs will experience a much smaller scattering and correspondingly, a much lower damping rate (Nature 2018, 562, 557.; Adv. Mater. 2018, 30, 1705318.; Sci. Adv. 2019, 5, eaav8690.). This will result in the high-Q resonances observed in the far field. A high-Q resonance can provide a strong light confinement at the nanoscale, and also is very important for the photonic applications of filter, switch, sensing, and electromagnetic radiation.

Secondly, in Ref. 50 (Nat. Commun. 2020, 11, 1158.), plasmon polaritons were discussed in patterned WTe_2 in the THz spectral range ($200 - 630 \text{ cm}^{-1}$: 16 to $50 \mu\text{m}$), with the in-plane hyperbolic regime mainly locating in a rather limited range from $435 - 630 \text{ cm}^{-1}$ ($16 - 23 \mu\text{m}$). This is due to that in-plane hyperbolic plasmon polaritons are contributed by the interplay of interband and intraband electron transitions in WTe_2 . When the excitation frequency is large, more interband transitions (from the deep valence band levels) will occur and overwhelm the intraband transitions. If the incident photon energies are too small for the electron interband transitions to take place, the plasmon polaritons will be mainly governed by the electron intraband transitions. Both effects will suppress the in-plane hyperbolicity and limit the operation frequency range. In the $\alpha\text{-MoO}_3$ polariton tuners reported in our study, the polaritons are originated from the PhPs, where the in-plane hyperbolicity exists in the Reststrahlen band with $\text{Re}(\epsilon_x) \cdot \text{Re}(\epsilon_y) < 0$. Because the Reststrahlen band is determined by the transverse and longitudinal optical phonon frequencies in $\alpha\text{-MoO}_3$, as shown in our study, the in-plane hyperbolicity of the polariton tuners can be observed in the THz range of $260 - 400 \text{ cm}^{-1}$ and LWIR range of $540 - 980 \text{ cm}^{-1}$. Such spectral range is wider than that of the patterned WTe_2 .

Thirdly, in Ref. 50 (Nat. Commun. 2020, 11, 1158.), the hyperbolic plasmon polaritons in patterned WTe_2 were measured at a rather low temperature of 10 K . Because the plasmon polaritons are contributed by the interband and intraband transitions of electrons, cryogenic cooling is required to suppress the lattice scattering of the electrons and thermal noises to guarantee accurate far-field spectroscopic characterizations. As shown by their results (Fig. 2b in Ref. 50), the polariton resonances will be broadened and suppressed against increase of temperature, which is not favorable for many photonic applications, especially for those operating at room temperature. In contrast, all of the far-field spectroscopic measurements of the $\alpha\text{-MoO}_3$ polariton tuners were conducted at room temperature. This is due to that the PhPs were contributed by coupling of photons and lattice vibrations, where the damping from the lattice scattering will be strongly suppressed (Nature 2018, 562, 557.; Adv. Mater. 2018, 30, 1705318.; Sci. Adv. 2019, 5, eaav8690.). This room-temperature operation manner enables practical photonic applications of the vdW polaritonic crystals, for example, as demonstrated in our study, the polarizing notch filters (PNFs).

Last but not least, the polariton resonances of the α -MoO₃ tuner is strongly dependent on the excitation polarization. By taking advantage of such characteristic, we constructed the PNF which combines a polarizer and a narrow band-rejection filter together into a single component, using the α -MoO₃ polariton tuner. Moreover, we showed that the rejection frequency of the PNFs could be tuned by changing the width of the ribbons. These tunable PNFs are able to block a selected monochromic laser with a given linear polarization, while passing light of all polarization states at wavelengths adjacent to the laser line. Such unique filters have broad application prospects in laser spectroscopy and optical communications, where the commercial products are rare, and especially, there is no commercial PNF in the LWIR and THz ranges.

To further clarify the novelty of our study in comparison to that of Ref. 50, the first paragraph of the section "Conclusions and prospects" has been modified as,

"We have successfully demonstrated direct far-field excitation and characterization of the tunable LWIR and THz HPhPs in biaxial vdW α -MoO₃ patterned into simple 1D-PRPs. The 1D-PRPs can act as polariton tuners that are sensitive to the excitation polarization and with light extinction ratios up to 6.5 dB and high Q-factors up to 300. Such a compositional set of output functions are tunable and strongly dependent upon the in-plane hyperbolic phonon polaritons in the α -MoO₃. It is noted that in comparison with the recently reported patterned vdW WTe₂ flakes with tunable in-plane hyperbolic plasmon polaritons at cryogenic conditions⁴⁵, the polariton tuners proposed in the current study may be more favorable for practical applications because of their room-temperature operation, broader spectral range, and much higher quality factors of the resonance modes."

Q2: The authors highlight advantages of ribbon spectroscopy compared to e.g. near-field microscopy studies. The discussion is not objective enough. For example, broadband infrared sources can be used and actually have been used. There are several papers published, e.g. ref. 5, 10, etc. Broadband spectra can be also obtained with EELS spectroscopy. The authors also do not consider that fabrication of ribbons may lead to damage of the ribbon edges that may give rise to additional damping and lowering of Q factors.

R2: We agree with the reviewer that the broadband polariton spectra can be obtained using EELS technique or near-field microscopy integrated with a broadband infrared light source. However, in EELS characterizations, the sample is usually milled down to ~ 10 nm for getting good signals. In particular, an ultrahigh vacuum sample condition is a prerequisite for the EELS. These two conditions make EELS characterizations require complex instrumentation and harsh sample preparations. In contrast, for the far-field polariton characterizations with the 1D ribbon array, the measurements can be readily conducted in a common FTIR spectrometer at ambient conditions.

The near-field microscopy integrated with a broadband infrared light source, *i.e.*, the infrared nanoscopy technique, has been indeed widely used in the studies of polaritonic materials such as graphene (Nano Lett. 2011, 11, 4701.), hBN (Science 2014, 343, 1125.), α -MoO₃ (Nature 2018, 562, 557.), and α -V₂O₅ (Nat. Mater. 2020, 19, 964.). The polaritonic dispersion relation and lifetime can be derived. However, the output of the infrared source in most commercially available s-SNOM system is restricted at the spectral range over 670 cm⁻¹ due to the limitation of current manufacturing technology, while the broadband polaritonic responses of α -MoO₃ resides in a spectral range down to 260 cm⁻¹. Although s-SNOM integrated with synchrotron infrared source is able to probe the polaritons in an ultrabroadband range from 440 – 1020 cm⁻¹ (ACS Nano 2022, 16, 3027.), it is not accessible to ordinary research group. Therefore, the commercially available broadband infrared source for infrared nanoscopy cannot cover the full Reststrahlen bands of α -MoO₃, making the characterization of PhPs in these bands difficult and challenge. Furthermore, the relatively weak power of the broadband infrared source at the edge frequency (*i.e.*, \sim 670 cm⁻¹) makes the signal-to-noise ratio of the spectrum weak, thereby affecting the accuracy and authenticity of the data. Particularly, a high-power broadband THz source remain challenge, making the near-field spectroscopic characterizations of THz polariton properties difficult. In contrast, in our far-field measurements, the light source integrated in the Bruker FTIR spectrometer (Vertex 70v) operates in a broad spectral range covering the LWIR and THz regimes, which guarantees the characterizations of all the HPhPs in the patterned α -MoO₃ flake. Another issue is the measurement time consumption. Whether it is EELS or infrared nanoscopy technology, it usually takes a long time to locate, focus, and measure the sample under the microscope (a transmission electron microscope for EELS and an atomic force microscope for infrared nanoscopy). Usually, it will take half an hour or more to collect the broadband spectrum of a typical sample. Contrarily, for far-field FTIR characterization, it only takes a few minutes for the same spectral measurement.

In addition to reveal the far-field spectroscopic characteristics of the polaritonic response of α -MoO₃, another main purpose of our study is to demonstrate the possible practical application of the HPhPs in α -MoO₃. Direct far-field excitation and characterization of the HPhPs can on hand directly obtain parameters related to the photonic applications, such as polarization characteristics, optical filtering characteristics, Q-factors, *etc.*, and on the other hand directly demonstrate the device application, just as the PNFs shown by our study.

In these regards, due to its broadband spectral range, facile operation, easy sample preparation, and high efficiency, the far-field spectroscopic characterizations of the HPhPs using the α -MoO₃ 1D ribbon arrays can greatly facilitate the study of the polaritonic properties.

As suggested by the reviewer, to further highlight the advantages of the far-field spectroscopy characterizations with the ribbon arrays, a discussion has been added in paragraph 2 in page 3 in the revised manuscript as,

"...It should be noted that usually three main techniques are employed for determining the

broadband polaritonic properties of 2D crystals, including the FTIR^{36,47,48}, electron energy loss spectroscopy (EELS)⁴⁹, and infrared nanoscopy^{4,9,28}. In comparison with the latter two techniques which usually require a complex instrumentation, harsh sample preparation, and time consumption, the far-field polariton characteristics of the 1D-PRPs can be readily measured in a common broadband FTIR spectrometer at ambient conditions, with low time-consuming, a high collection efficiency, and over a large sample area."

We also agree with the reviewer that the fabrication of ribbons can lead to damage of the ribbon edges, which can bring in additional damping and lowering of Q factors. Usually this issue is inevitable during patterning of the vdW crystal for various characterizations and device applications. To evaluate the polariton damping in the α -MoO₃ 1D ribbon arrays, we extract the lifetime of HPhPs in the array structures by fitting the reflection spectra with a Fano line-shape function (Fig. R1a, Eq. R1). As shown in Fig. R1c, the polariton lifetime of the 1D ribbon arrays ranges from 0.2 ps to 3.0 ps for Reststrahlen Band 1 to Band 3.

We also performed near-field nano-imaging measurements on the same α -MoO₃ flake in the unpatterned region. It should be noted that due to the limitation of the quantum cascade lasers integrated into our s-SNOM system, the near-field measurements can only be conducted in the frequency range of 890 to 1250 cm⁻¹ (Band 2 and 3). In addition, to make a direct comparison, the excitation frequency of the s-SNOM was set close to the resonance frequency of the 1D ribbon arrays measured in the far-field. Specifically, for an arrays with the ribbon longitudinal axis pointing along [100] ([001]) direction, the resonance frequency was measured from far-field as ω_0 . The same frequency ω_0 was then employed as the excitation frequency for the near-field measurements. Consequently, clear interference fringes can be observed along the [001] ([100]) crystallographic direction (Fig. R1b). According to the line profiles extracted from the near-field interference fringes (Fig. R1b, insets), it is able to obtain the propagation length of the HPhPs upon an excitation frequency of ω_0 . Specifically, a typical line profile along [100] crystallographic direction of α -MoO₃ can be fitted using the equation (Nature 2018, 562, 557.),

$$y = Ax^{-0.5} e^{-\frac{x}{t_0}} \sin\left(\pi \frac{x-x_c}{w}\right), \quad A > 0, \quad w > 0, \quad t_0 > 0. \quad (\text{R3})$$

where t_0 denotes the polariton propagation length. The group velocity v_g of the HPhPs in α -MoO₃, which is defined as $v_g = \partial\omega/\partial k$, can be extracted from the polariton dispersion relation calculated using the analytical model (Note S6 in the Supplementary Information part of the revised manuscript). Afterwards, the lifetime of the HPhPs in the unpatterned α -MoO₃ can be calculated as $\tau = L/v_g$.

As shown in Fig. R1c, in comparison with the unpatterned α -MoO₃, the phonon polariton lifetime in the ribbon arrays are reduced by 20%–57% (Table R1). Such lifetime reduction is attributed to additional scattering by the rough edges of the ribbons or impurities introduced during the fabrication process. Despite the lifetime reduction, the 1D ribbon arrays still exhibit high Q factors upto 300. The lifetime of HPhPs in the α -MoO₃ ribbon arrays can be further

improved by optimizing the manufacturing parameters.

As suggested by the reviewer, a discussion on the polariton damping induced by fabrication of the ribbons is added in paragraph 3 in page 8 as,

"The far-field spectra also allow for extracting the polariton lifetimes, which span from 0.2 to 3.0 ps in the three Reststrahlen bands (Supplementary Note S5, Fig. S9, and Table S1). It is noted that fabrication of 1D-PRPs can lead to damage of the ribbon edges, which can reduce the polariton lifetimes and Q factors. Usually this issue is inevitable during patterning of the vdW crystal for various characterizations and device applications. To evaluate the additional polariton damping induced by the patterning processes, we performed near-field measurements on the same α -MoO₃ flake in the unpatterned region and extracted the intrinsic polariton lifetimes (Supplementary Note S5, Fig. S9, and Table S1). In comparison with the unpatterned α -MoO₃, the phonon polariton lifetime in the ribbon arrays are reduced by 20%–57%. Such lifetime reduction is attributed to additional scattering by the rough edges of the ribbons or impurities introduced during the fabrication process. Despite the lifetime reduction, the 1D-PRPs still exhibit high Q factors upto 300. Although such a value is smaller than that of resonances sustained by an unpatterned and naturally grown α -MoO₃ ribbon⁶¹, it can be further improved by optimizing the processing parameters."

In addition, the detailed discussion on extraction of the polariton lifetimes from far-field and near-field measurements has been added as Note S5 in Supplementary Information part as,

"Supplementary Note S5. Extraction and comparison of the polariton lifetimes obtained from far-field and near-field measurements

We extracted the lifetime of HPhPs in the 1D-PRPs by fitting the reflection spectra with the Fano line-shape function (Eq. S1 and Fig. S9a). The lifetime τ at a resonance frequency ω_0 can be derived as $\tau = 2\hbar/\Gamma$. As shown in Fig. S9c, the polariton lifetime of the 1D-PRPs ranges from 0.5 ps to 3.0 ps for Reststrahlen Band 1 to Band 3. We also performed near-field nano-imaging measurements on the same α -MoO₃ flake in the unpatterned region. A scattering type scanning near-field optical microscope was employed for the near-field characterizations (neaSNOM, neaspec GmbH)^{6,7}. It should be noted that due to the limitation of the quantum cascade lasers integrated into our s-SNOM system, the near-field measurements can only be conducted in the frequency range of 890 to 1250 cm⁻¹ (Band 2 and 3). In addition, to make a direct comparison, the excitation frequency of the s-SNOM was set close to the resonance frequency of the 1D-PRPs measured in the far-field. Specifically, for an 1D-PRP with the ribbon longitudinal axis pointing along [100] ([001]) direction, the resonance frequency was measured from far-field as ω_0 . The same frequency ω_0 was then employed as the excitation frequency for the near-field measurements. Consequently, clear interference fringes can be observed along the [001] ([100]) crystallographic direction (Fig. S9b). According to the line profiles extracted from the near-field interference fringes (Fig. S9b, insets), it is able to obtain the propagation length of the HPhPs upon an excitation frequency of ω_0 . A typical line profile along [100] crystallographic direction of α -MoO₃ can be

fitted using the equation⁸,

$$y = Ax^{-0.5} e^{-\frac{x}{t_0}} \sin\left(\pi \frac{x-x_c}{w}\right), \quad A > 0, \quad w > 0, \quad t_0 > 0. \quad (\text{S13})$$

where t_0 denotes the polariton propagation length. The group velocity v_g of the HPhPs in α -MoO₃, which is defined as $v_g = \partial\omega/\partial k$, can be extracted from the polariton dispersion relation calculated using the analytical model (see Note S6 below). Afterwards, the lifetime of the HPhPs in the unpatterned α -MoO₃ can be calculated as $\tau = L/v_g$.

The comparison of the polariton lifetimes obtained from far-field and near-field measurements is shown in Fig. S9c and Table S1."

Fig. R1 are added as Fig. S9 in the Supplementary Information part. Table R1 is added as Table S1 in the Supplementary Information part.

Q3: *The authors introduce a reciprocal lattice momentum G , but what is meant by “lattice”. Do the ribbon arrays have as a lattice, i.e. collectively (e.g. supporting lattice resonances) or do they couple to each other. It seems that they behave as individual ribbons and thus the words lattice, diffract, etc. are confusing/misleading. The authors also cite ref. 59 that is even more confusing because the phase matching condition is given by the period of the grating and not by the ribbon width.*

R3: To excite the HPhPs from far-field, it is required to compensate the large wavevector mismatch between free-space photons (k_0) and polaritons (q_{PhPs}). In our study, this can be achieved from two pathways (Fig. R5). Specifically, the first one is formation of Fabry–Pérot resonance (FPR). When the ribbon arrays are illuminated at normal incidence with an electric field pointing perpendicularly to the ribbon longitudinal axis, the ribbon edges will act as subwavelength-scale structures providing evanescent fields with high momentum. The evanescent waves can then hybridize with the optical phonons in α -MoO₃ and excite HPhPs propagating transverse to the ribbons. Once the polaritons are stimulated, they will form standing-wave resonances, i.e., the FPR, by multiple reflections from the ribbon edges under the condition,

$$q_{\text{PhPs}}w + \Phi = \pm m\pi, \quad m = 1, 2, 3, \dots \quad (\text{R4})$$

where w is the ribbon width, Φ is the possible phase shift upon reflection at the edges. The second one is that the 1D ribbon array acts as a grating structure. The incident waves (normal incidence) will be scattered into guided waves (GWs) propagating transverse to the ribbons if the following condition is satisfied (Appl. Opt. 1993, 32, 2606.),

$$k\Lambda = \pm n2\pi, \quad n = 1, 2, 3, \dots \quad (\text{R5})$$

where k and Λ are the wavevector of the GWs and period of the ribbon array, respectively. The GWs can then couple with the α -MoO₃ ribbons and convert into polaritons with the same wavevectors (ACS Nano 2012, 6, 7806.). Considering that in our study the Λ was deliberately set as $2w$, Eq. R4 can be written as,

$$q_{\text{PhPs}w} = \pm n\pi, n = 1, 2, 3, \dots \quad (\text{R6})$$

The Eq. R4 and Eq. R6 are very close to each other except for the phase difference Φ in Eq. R4. To ascertain which pathway of the two dominates the excitation of the HPhPs resonances, the energy absorbed by an isolated individual ribbon (labeled as Ribbon 1) and a typical single ribbon (labeled as Ribbon 2) in the 1D ribbon array is compared. To that end, the energy consumption and electric field distributions within the Ribbon 1 and Ribbon 2 are calculated. Both ribbons have thicknesses of 200 nm and w of 800 nm. The longitudinal axes of the ribbons are set along [100] and [001] crystallographic directions, respectively. The illumination power is kept the same. As shown in Fig. R6a, the energy absorption in Ribbon 1 and Ribbon 2 is both frequency dependent, with the maxima at $\omega_{01} = 878 \text{ cm}^{-1}$ and $\omega_{02} = 876 \text{ cm}^{-1}$, respectively. The ω_{01} and ω_{02} are very close with each other and coincide with the polariton resonance frequency of the 1D ribbon array extracted from the far-field reflectance spectrum (Fig. R6a, dashed lines). The absorption maximum in Ribbon 2 is an order of magnitude larger than that in Ribbon 1. Moreover, the electric field distribution inside the Ribbon 1 is much smaller than that in the Ribbon 2 (Fig. R6b). Similar situation can be observed for ribbons orientating along the [001] crystallographic direction (Fig. R6c and R6d). From these analyses it can be deduced that the excitation of HPhPs are originated from the synergy between GWs of the array and FPR in an individual ribbon: the scattering of light at the ribbon edges excite the polariton FPR, while the GWs further couple with and transfer energy to the polariton FPRs.

Due to its periodicity, in our study we define the 1D ribbon array as a "1D lattice", giving rise to the reciprocal lattice momentum G . Because the period of the ribbon array is $\Lambda = 2w$, the G then equals to $m\pi/w$. This is why the phase matching condition appears to be defined by the ribbon width rather than the grating period.

Figure R5. Schematic showing excitation of polariton resonances in the 1D-PRP from far-field. The incident waves will be scattered by the sharp ribbon edges into evanescent waves with large momenta, whereby HPhPs propagating transverse to the ribbons are excited. FPRs can then be formed upon the multiple polariton reflections from the ribbon edges. Simultaneously, the 1D-PRPs can also diffract the incident light into guided waves propagating perpendicular to the ribbon long axis, whose wavevectors are much larger than the free-space waves. These guided waves can then couple with and transfer energy to the polariton FPRs.

Figure R6. (a, c) Comparison of the simulated energy absorption by an individual ribbon (Ribbon 1: black lines) and a typical single ribbon (Ribbon 2: red lines) in the 1D ribbon array. The calculated far-field reflectance spectra are included for reference (dashed red lines). The longitudinal axes of the ribbons are parallel to [001] (a) and [100] (c) crystallographic axis of α - MoO_3 , respectively. (b, d) Optical near-field distributions of Ribbon 1 (lower) and Ribbon 2 (upper). The near-field distributions are drawn on the cross section perpendicular to the ribbon transverse axis, *i.e.*, the $x-z$ plane for (b) and $y-z$ plane for (d).

To clarify the excitation mechanisms of the PhPs in the 1D ribbon array as well as the phase matching condition, the discussion in paragraph 2 in page 4 is modified as,

" A tuner comprised of 1D-PRPs (Fig. 1c and Supplementary Fig. S2) is able to overcome the large momentum mismatch and excite the HPhPs^{52,53}. The incident waves will be scattered by the

sharp ribbon edges into evanescent waves with large momenta, whereby HPhPs propagating transverse to the ribbons are excited. Fabry–Pérot resonances (FPRs) can then be formed upon the multiple polariton reflections from the ribbon edges. Simultaneously, the 1D-PRPs can also diffract the incident light into guided waves propagating perpendicular to the ribbon long axis, whose wavevectors are much larger than the free-space waves⁵⁴. These guided waves can then couple with and transfer energy to the polariton FPRs (see Supplementary Note S2, Fig. S5, and Fig. S6 for more discussion on the excitation of HPhPs by the 1D-PRP). The polariton energy will be dissipated by the lattice vibrations or radiated back to the free space, as manifested by resonance peaks and valleys in the corresponding reflectance and transmission spectra, respectively."

A discussion has been also added in paragraph 2 in page 7 as,

" The evolution of the HPhP resonances with changing w can be understood by considering that the excitation of HPhPs are originated from the synergy between guided waves of the array and FPRs in an individual ribbon: the scattering of light at the ribbon edges excite the polariton FPR, while the guided waves further couple with and transfer energy to the polariton waves (see Supplementary Note S2, Fig. S5, and Fig. S6). The conditions for occurrence of the FPRs and guided waves are $q_{\text{PhPs}}w = \pm m\pi$ and $q_{\text{PhPs}}\Lambda = \pm n2\pi$, respectively, with $m, n = 1, 2, 3, \dots$ ^{45,57}. Because in our study the Λ is deliberately set as $2w$, these two equations are the same. Therefore, each w corresponds to an in-plane polariton wavevector of $q_{\text{PhPs}} = \mp \frac{m\pi}{w}$. When w changes the resonance peak will scale according to the in-plane polariton dispersion relations, $\omega(q_{\text{PhPs}})\dots$ "

In addition, a discussion has been added in Supplementary Information as Note S2,

"Supplementary Note S2. Discussion on the excitation mechanism of HPhPs by the 1D-PRP

To excite the HPhPs from far-field, it is required to compensate the large wavevector mismatch between free-space photons (k_0) and polaritons (q_{PhPs}). In our study, this can be achieved from two pathways (Fig. S5). Specifically, the first one is formation of Fabry–Pérot resonance (FPR). When the ribbon arrays are illuminated at normal incidence with an electric field pointing perpendicularly to the ribbon longitudinal axis, the ribbon edges will act as subwavelength-scale structures providing evanescent fields with high momentum. The evanescent waves can then hybridize with the optical phonons in $\alpha\text{-MoO}_3$ and excite HPhPs propagating transverse to the ribbons. Once the polaritons are stimulated, they will form standing-wave resonances, *i.e.*, the FPR, by multiple reflections from the ribbon edges under the condition,

$$q_{\text{PhPs}}w + \Phi = \pm m\pi, m = 1, 2, 3, \dots \quad (\text{S2})$$

where w is the ribbon width, Φ is the possible phase shift upon reflection at the edges. The second one is that the 1D-PRP acts as a grating structure. The incident waves will be scattered into guided waves (GWs) propagating transverse to the ribbons if the following condition is satisfied³,

$$k\Lambda = \pm n2\pi, n = 1, 2, 3, \dots \quad (\text{S3})$$

where k and Λ are the wavevector of the GWs and period of the ribbon array, respectively. The GWs can then couple with the α -MoO₃ ribbons and convert into HPhPs with the same wavevectors⁴. Considering that in our study the Λ was deliberately set as $2w$, Eq. S3 can be written as,

$$q_{\text{HPhPs}}w = \pm n\pi, n = 1, 2, 3, \dots \quad (\text{S4})$$

The Eq. S2 and Eq. S4 are very close to each other except for the phase difference Φ in Eq. S24. To ascertain which pathway of the two dominates the excitation of the HPhPs resonances, the energy absorbed by an individual ribbon (labeled as Ribbon 1) and a typical single ribbon (labeled as Ribbon 2) in the 1D ribbon array is compared. To that end, the energy consumption and electric field distributions within the Ribbon 1 and Ribbon 2 are calculated. Both ribbons have thicknesses of 200 nm and w of 800 nm. The longitudinal axes of the ribbons are set along [100] and [001] crystallographic directions, respectively. The illumination power is kept the same. As shown in Fig. S6a, the energy absorption in Ribbon 1 and Ribbon 2 is both frequency dependent, with the maxima at $\omega_{01} = 878 \text{ cm}^{-1}$ and $\omega_{02} = 876 \text{ cm}^{-1}$, respectively. The ω_{01} and ω_{02} are very close with each other and coincide with the polariton resonance frequency of the 1D ribbon array extracted from the far-field reflectance spectrum (Fig. S6a, dashed lines). The absorption maximum in Ribbon 2 is an order of magnitude larger than that in Ribbon 1. Moreover, the electric field distribution inside the Ribbon 1 is much smaller than that in the Ribbon 2 (Fig. S6b). Similar situation can be observed for ribbons orientating along the [001] crystallographic direction (Fig. S6c and S6d). From these analyses it can be deduced that the excitation of HPhPs are originated from the synergy between GWs of the array and FPR in an individual ribbon: the scattering of light at the ribbon edges excite the polariton FPR, while the GWs further couple with and transfer energy to the polariton waves.

Due to its periodicity, in our study we define the 1D-PRP as a "1D lattice", giving rise to the reciprocal lattice momentum G . Because the period of the ribbon array is $\Lambda = 2w$, the G then equals to $m\pi/w$.

Supplementary Fig. S5 | Schematic showing excitation of polariton resonances in the 1D-PRP from far-field. The incident waves will be scattered by the sharp ribbon edges into

evanescent waves with large momenta, whereby HPhPs propagating transverse to the ribbons are excited. Fabry–Pérot (FP) resonances can then be formed upon the multiple polariton reflections from the ribbon edges. Simultaneously, the 1D-PRPs can also diffract the incident light into guided waves propagating perpendicular to the ribbon long axis, whose wavevectors are much larger than the free-space waves. These guided waves can then couple with and transfer energy to the polariton FP resonances.

Supplementary Fig. S6 | Comparison of the energy absorption by an individual ribbon and a typical single ribbon in the 1D ribbon array. **a, c** Simulated energy absorption by an individual ribbon (Ribbon 1: black lines) and a typical single ribbon (Ribbon 2: red lines) in the 1D ribbon array. The calculated far-field reflectance spectra are included for reference (dashed red lines). The longitudinal axes of the ribbons are parallel to [001] (**a**) and [100] (**c**) crystallographic axis of α -MoO₃, respectively. **b, d** Optical near-field distributions of Ribbon 1 (lower) and Ribbon 2 (upper). The near-field distributions are drawn on the cross section perpendicular to the ribbon transverse axis, *i.e.*, the x - z plane for (**b**) and y - z plane for (**d**)."

Fig. R5 and Fig. R6 are added as Fig. S5 and Fig. S6 in the Supplementary Information. The two references "Appl. Opt. 1993, 32, 2606." and "ACS Nano 2012, 6, 7806." are added as Ref. 3 and 4 in the Reference list of Supplementary Information. The reference "Appl. Opt. 1993, 32, 2606." has also been added as Ref. 54 in the Reference list.

Q4: *The authors write that the polariton propagate along the ribbons and form standing-wave resonances due to reflection at ribbon edges. It seems that that the polaritons propagate transverse to the ribbons, *i.e.* across the ribbons rather than along the ribbons. I would say the*

authors measure Farby-Perot modes across the ribbons but they do not clearly explain this. The polaritons also may be phase shifted upon reflection at the edges, which needs to be considered in the analysis.

R4: We apologize for writing that "the polariton propagate along the ribbons and form standing-wave resonances due to reflection at ribbon edges". This is a mistake, we actually meant to say that the standing-wave resonances are formed by polaritons propagating transversely to the ribbons being reflected at the edges. We would like to explain more on the mechanisms for far-field excitation of the HPhPs.

To excite the HPhPs from far-field, it is required to compensate the large wavevector mismatch between free-space photons (k_0) and polaritons (q_{PhPs}). In our study, this can be achieved from two pathways. Specifically, the first one is formation of Fabry–Pérot resonance (FPR). When the ribbon arrays are illuminated at normal incidence with an electric field pointing perpendicularly to the ribbon longitudinal axis, the ribbon edges will act as subwavelength-scale structures providing evanescent fields with high momentum. The evanescent waves can then hybridize with the optical phonons in α -MoO₃ and excite HPhPs propagating transverse to the ribbons. Once the polaritons are stimulated, they will form standing-wave resonances, *i.e.*, the FPR, by multiple reflections from the ribbon edges under the condition,

$$q_{\text{PhPs}}w + \Phi = m\pi, m = 1, 2, 3, \dots \quad (\text{R7})$$

where w is the ribbon width, Φ is the possible phase shift upon reflection at the edges. The second one is that the 1D ribbon array acts as a grating structure. The incident waves (normal incidence) will be scattered into guided waves (GWs) propagating transverse to the ribbons if the following condition is satisfied (Appl. Opt. 1993, 32, 2606.),

$$k\Lambda = n2\pi, n = 1, 2, 3, \dots \quad (\text{R8})$$

where k and Λ are the wavevector of the GWs and period of the ribbon array, respectively. The GWs can then couple with the α -MoO₃ ribbons and convert into HPhPs with the same wavevectors (ACS Nano 2012, 6, 7806.). Considering that in our study the Λ was deliberately set as $2w$, Eq. R4 can be written as,

$$q_{\text{PhPs}}w = n\pi, n = 1, 2, 3, \dots \quad (\text{R9})$$

The Eq. R7 and Eq. R9 are very close to each other except the phase difference Φ in Eq. R7. To ascertain which pathway of the two dominates the excitation of the HPhPs resonances, the energy absorbed by an isolated individual ribbon (labeled as Ribbon 1) and a typical single ribbon (labeled as Ribbon 2) in the 1D ribbon array is compared. To that end, the energy consumption and electric field distributions within the Ribbon 1 and Ribbon 2 are calculated. Both ribbons have thicknesses of 200 nm and w of 800 nm. The longitudinal axes of the ribbons

are set along [100] crystallographic direction. The illumination power is kept the same. As shown in Fig. R6a, the energy absorption in Ribbon 1 and Ribbon 2 is both frequency dependent, with the maxima at $\omega_{01} = 878 \text{ cm}^{-1}$ and $\omega_{02} = 876 \text{ cm}^{-1}$, respectively. The absorption maximum in Ribbon 2 is an order of magnitude larger than that in Ribbon 1. The ω_{02} coincides with the polariton resonance frequency of the 1D ribbon array extracted from the far-field reflectance spectrum (Fig. R6a, dashed line). Moreover, the electric field distribution inside the Ribbon 1 is much smaller than that in the Ribbon 2 (Fig. R6b). Similar situation can be observed for ribbons orientating along the [001] crystallographic direction (Fig. R6c and R6d). From these analyses it can be deduced that the excitation of HPhPs are originated from the synergy between GWs of the array and FPR in an individual ribbon: the scattering of light at the ribbon edges excite the polariton FPR, while the GWs further couple with and transfer energy to the polariton FPRs.

As raised by the reviewer, reflection of polariton waves by the ribbon edges may induce phase shifts, which can violate the condition for the FPRs by a phase of 2Φ (Phys. Rev. B 2014, 90, 041407(R)). A previous theoretical result showed that in a monolayer graphene, the Φ for plasmon polaritons is $\sim 0.75\pi$ (Phys. Rev. B 2014, 90, 041407(R)). In comparison with the plasmon polariton in graphene, the propagation and reflection of the in-plane HPhPs are rather complicated, making the phase shift difficult to be predicted. In our analyses, the Φ is taken as π according to a very recent study (2022, <https://arxiv.org/ftp/arxiv/papers/2206/2206.14886.pdf>). The good agreement between the experimental measurements and calculated results further validate our setting.

We thank very much the reviewer for pointing out such an interesting issue. The phase shifts of the in-plane HPhPs upon reflected by the edges will be further studied in our future work.

To clarify the excitation mechanisms of the HPhPs from far-field and the possible phase shifts of the HPhPs, a discussion is added in paragraph 1 in page 8 as,

"...It is noted that reflection of polariton waves by the ribbon edges may induce phase shifts, which can violate the condition for the FPRs by a phase of 2Φ ⁵⁸. A previous theoretical result showed that in a monolayer graphene, the Φ for plasmon polaritons is $\sim 0.75\pi$ ⁵⁸. In comparison with the plasmon polariton in graphene, the propagation and reflection of the in-plane HPhPs are rather complicated, making the phase shift difficult to be predicted. In our analyses, the Φ is taken as π according to a very recent study⁵⁹. The good agreement between the experimental measurements and calculated results further validate our setting."

A discussion on the far-field excitation mechanisms of the HPhPs in 1D ribbon array has been added in Supplementary Information as Note S2. The two references "Phys. Rev. B 2014, 90, 041407(R)." and "2022, <https://arxiv.org/ftp/arxiv/papers/2206/2206.14886.pdf> " are added as Ref. 58 and 59 in the reference list.

Q5: *It is often not clear whether it is simply the high Q factor of the resonances or the anisotropy of the material that is important. For example, the authors mention tunable notch filters in Fig. 5. They show narrow resonances, which spectrally shift when the ribbon width is kept same but the orientation is rotated relative to crystal axes. This is interesting, but narrow resonances can be also shifted with ribbons made out of an isotropic material (where high Q factor polaritons exist) by changing the ribbon width. I am wondering what is the specific advantage of using anisotropic materials such as MoO_3 , apart of the high Q factor that MoO_3 has (which is not necessarily a consequence of its anisotropy).*

R5: We agree with the reviewer that the high- Q polariton resonances can also be induced in ribbon arrays made out of an in-plane isotropic polaritonic vdW crystal. It is indeed that the frequency of such resonances can also be tuned by changing the ribbon width (For example, hBN ribbons reported in Light Sci. Appl. 2018, 7, e17172.). We think the polariton resonances sustained by the 1D ribbon arrays of α - MoO_3 , which exhibits an in-plane hyperbolicity, have two advantages.

First, for ribbon arrays fabricated on an in-plane isotropic vdW crystal of a fixed thickness, the resonance frequency can only be tuned by changing the ribbon width w . Take hBN for example, to increase a resonance with higher frequency, the ribbons need to be narrowed to a few tens of nanometers or even smaller (Light Sci. Appl. 2018, 7, e17172.). This is not only a challenge for nowadays lithography technique, but also brings in additional damping for the resonance because defects and impurities are more easily to be accumulated onto the thinner ribbons during the fabrication processes. However, such an issue will be alleviated for ribbon arrays fabricated with α - MoO_3 . The in-plane hyperbolicity of α - MoO_3 makes it possible to tune the resonance frequency of the 1D ribbon array by rotating the ribbon long axis with respect to the [001] crystallographic axis while fixing the ribbon width. This feature can greatly simplify the fabrication processes of ribbon arrays of different resonance frequencies without bringing additional losses.

Secondly, in Reststrahlen Band 1 and Band 2, the polariton isofrequency contour (IFC) in the basal plane of the α - MoO_3 is a hyperbola. For a polariton mode with wavevector approaching the asymptote of the hyperbola, the polariton momentum will become remarkably high. This will generate a much stronger electromagnetic field confinement than the polariton modes with wavevectors away from the asymptote. These modes can significantly enhance light-matter interactions at nanoscale and lead to various applications, such as enhanced light emission, ultrasensitive biosensing, and nonlinear optical signal generations. For these applications usually a fixed operation frequency is preferred. In the ribbon arrays made out of α - MoO_3 , tuning the orientation and width of the ribbons can both tailor the resonance frequency. Therefore, it is possible to induce a high-momentum polariton mode while fixing its resonance frequency by simultaneously orientating the ribbon long axis along the asymptote of the hyperbolic IFC and tuning the ribbon width (please refer to Fig. 3d–f in our original manuscript). This feature can

open a new avenue for the applications just mentioned.

To clarify the specific advantage of fabricating ribbon arrays using the anisotropic α -MoO₃, a discussion has been added in paragraph 2 in page 10 as,

"High-Q resonances can also be induced in 1D-PRPs made out of an in-plane isotropic polaritonic vdW crystal⁴⁴. The frequency of such resonances can also be tuned by changing the ribbon width, but the α -MoO₃ 1D-PRPs proposed in the current study is unique. The in-plane hyperbolicity of α -MoO₃ makes it possible to tune the resonance frequency of the 1D-PRPs by rotating the ribbons while fixing their widths. This feature can greatly simplify the fabrication processes of arrays with different resonance frequencies, and small additional damping will be introduced because the ribbon width is unchanged. Moreover, for a polariton mode with wavevector approaching the asymptote of the hyperbolic IFC, the polariton momentum will become remarkably high. This will generate a much stronger electromagnetic field confinement than those with wavevectors away from the asymptote. These modes can significantly enhance light-matter interactions at nanoscale and lead to various applications, such as enhanced light emission, ultrasensitive biosensing, and nonlinear optical signal generations. For these applications usually a fixed operation frequency is preferred. In the 1D-PRPs made out of α -MoO₃, tuning the orientation and width of the ribbons can both tailor the resonance frequency. Therefore, it is possible to induce a high-momentum polariton mode while fixing its resonance frequency by simultaneously orientating the ribbon long axis along the asymptote of the hyperbolic IFC and tuning the ribbon width. This feature can open a new avenue for the applications just mentioned."

Q6: Since the authors claim that they report a first far field characterization, a more detailed discussion of the linewidths/lifetimes could be provided, particularly discussing whether fabrication uncertainties have some influence compared to lifetimes obtained from near-field characterization.

R6: As suggested by the reviewer, to further discuss the lifetimes of PhPs in our study, we carried out more quantitative analysis on the far-field spectra of the pattern structures of α -MoO₃. Specifically, we extract the lifetime of HPhPs in the array structures by fitting the reflection spectra with a Fano line-shape function (Eq. R1 and Fig. R1a, and please see more details in response to Q2 of Reviewer #3). As shown in Fig. R7, the lifetimes for the HPhPs measured from the 1D ribbon array range from 0.2 ps to 3.0 ps in the three Reststrahlen bands.

We also performed near-field nano-imaging measurements on the same α -MoO₃ flake in the unpatterned region. According to the line profiles extracted from the near-field interference fringes (Fig. R1b, insets), it is able to obtain the propagation length of the HPhPs and thereafter the polariton lifetime upon an excitation frequency of ω_0 (please see more details in response to

Q2 of Reviewer #3). As shown in Fig. R7, in comparison with the lifetime obtained from the unpatterned α -MoO₃, the polariton lifetime in the ribbon arrays are reduced by 20%–57% (Table R1). Such lifetime reduction is attributed to additional scattering by the rough edges of the ribbons or impurities introduced during the fabrication process. Despite the lifetime reduction, the 1D ribbon arrays still exhibit high Q factors upto 300. The lifetime of HPhPs in the α -MoO₃ ribbon arrays can be further improved by optimizing the manufacturing parameters.

Figure R7. Dependence of polariton lifetimes on resonance frequency ω_0 of the 1D ribbon arrays. The solid spheres are extracted from the far-field spectroscopic measurements, while the open symbols are results from near-field characterizations. For far-field measurements, the long axes of the ribbons are paralleled to the [100] (red) and [001] (blue) crystallographic directions, respectively. For near-field measurements, the line profiles are analyzed along [001] (red) and [100] (blue) crystallographic directions, respectively.

As suggested by the reviewer, a discussion on the lifetimes of HPhPs has been added in paragraph 3 in page 8 as,

"The far-field spectra also allow for extracting the polariton lifetimes, which span from 0.2 to 3.0 ps in the three Reststrahlen bands (Supplementary Note S5, Fig. S9, and Table S1). It is noted that fabrication of 1D-PRPs can lead to damage of the ribbon edges, which can reduce the polariton lifetimes and Q factors. Usually this issue is inevitable during patterning of the vdW crystal for various characterizations and device applications. To evaluate the additional polariton damping induced by the patterning processes, we performed near-field measurements on the same α -MoO₃ flake in the unpatterned region and extracted the intrinsic polariton lifetimes (Supplementary Note S5, Fig. S9, and Table S1). In comparison with the unpatterned α -MoO₃, the phonon polariton lifetime in the ribbon arrays are reduced by 20%–57%. Such lifetime reduction is attributed to additional scattering by the rough edges of the ribbons or impurities introduced during the fabrication process. Despite the lifetime reduction, the 1D-PRPs still exhibit high Q factors upto 300. Although such a value is smaller than that of resonances

sustained by an unpatterned and naturally grown α -MoO₃ ribbon⁶¹, it can be further improved by optimizing the processing parameters."

In addition, the detailed discussion on extraction of the polariton lifetimes from far-field and near-field measurements has been added as Note S5 in Supplementary Information part as,

"Supplementary Note S5. Extraction and comparison of the polariton lifetimes obtained from far-field and near-field measurements

We extracted the lifetime of HPhPs in the 1D-PRPs by fitting the reflection spectra with the Fano line-shape function (Eq. S1 and Fig. S9a). The lifetime τ at a resonance frequency ω_0 can be derived as $\tau = 2\hbar/\Gamma$. As shown in Fig. S9c, the polariton lifetime of the 1D-PRPs ranges from 0.2 ps to 3.0 ps for Reststrahlen Band 1 to Band 3. We also performed near-field nano-imaging measurements on the same α -MoO₃ flake in the unpatterned region. A scattering type scanning near-field optical microscope was employed for the near-field characterizations (neaSNOM, neaspec GmbH)^{6,7}. It should be noted that due to the limitation of the quantum cascade lasers integrated into our s-SNOM system, the near-field measurements can only be conducted in the frequency range of 890 to 1250 cm⁻¹ (Band 2 and 3). In addition, to make a direct comparison, the excitation frequency of the s-SNOM was set close to the resonance frequency of the 1D-PRPs measured in the far-field. Specifically, for an 1D-PRP with the ribbon longitudinal axis pointing along [100] ([001]) direction, the resonance frequency was measured from far-field as ω_0 . The same frequency ω_0 was then employed as the excitation frequency for the near-field measurements. Consequently, clear interference fringes can be observed along the [001] ([100]) crystallographic direction (Fig. S9b). According to the line profiles extracted from the near-field interference fringes (Fig. S9b, insets), it is able to obtain the propagation length of the HPhPs upon an excitation frequency of ω_0 . A typical line profile along [100] crystallographic direction of α -MoO₃ can be fitted using the equation⁸,

$$y = Ax^{-0.5} e^{-\frac{x}{t_0}} \sin\left(\pi \frac{x-x_c}{w}\right), \quad A > 0, \quad w > 0, \quad t_0 > 0. \quad (\text{S13})$$

where t_0 denotes the polariton propagation length. The group velocity v_g of the HPhPs in α -MoO₃, which is defined as $v_g = \partial\omega/\partial k$, can be extracted from the polariton dispersion relation calculated using the analytical model (see Note S6 below). Afterwards, the lifetime of the HPhPs in the unpatterned α -MoO₃ can be calculated as $\tau = L/v_g$.

The comparison of the polariton lifetimes obtained from far-field and near-field measurements is shown in Fig. S9c and Table S1."

Fig. R1 are added as Fig. S9 in the Supplementary Information part. Table R1 is added as Table S1 in the Supplementary Information part.

REVIEWERS' COMMENTS

Reviewer #1 (Remarks to the Author):

As the authors stated, it is widely known that one can induce polaritonic resonances with far-field excitation by patterning a polaritonic material into an array of nanoresonators, but this work applies the same technique to in-plane anisotropic HPhPs in α -MoO₃. Considering that the orientation-dependent polariton dispersion of α -MoO₃ is also well known, I still feel that this work is an expected extension of the previous works and would thus be unlikely to attract interest from a wide range of researchers. Therefore, I recommend the authors consider submitting this article to a more specialized journal.

Reviewer #2 (Remarks to the Author):

The authors addressed all the comments raised during the earlier review. The manuscript quality is substantially improved with the revisions suggested by the reviewers. The reported study is significant for the 2D materials community and IR region optoelectronics applications. I have only one comment to be addressed by the authors. I believe the manuscript could be accepted for publication after addressing the following comment.

The authors claim that the linewidth of the phonon modes is modified on structuring the MoO₃ crystal. The authors estimated the lifetime of the polariton modes for non-patterned MoO₃ and ribbon arrays and reported a 20%–57% reduction to support their claim. I believe, this estimation is cumulative effect over the modified (edges) and unmodified regions. The authors modified the dielectric permittivity of the medium to fit the polariton modes' linewidths. However, the modification in the dielectric permittivity won't be uniform over the whole width of the grating. The accurate estimation of the phonon linewidth can only be seen by the spatially resolved Raman scattering measurements. I suggested the authors to report the phonon linewidth modification from the spatially resolved Raman measurements. This information could be useful for fundamental studies and the technological development of the 2D material for photonic and optoelectronic devices development.

Reviewer #3 (Remarks to the Author):

The authors clarified the points raised by the reviewer and the manuscript can be accepted.

Response to Reviewer #2

We thank very much this reviewer for the concerns and valuable suggestions on our manuscript.

Comment: The authors addressed all the comments raised during the earlier review. The manuscript quality is substantially improved with the revisions suggested by the reviewers. The reported study is significant for the 2D materials community and IR region optoelectronics applications. I have only one comment to be addressed by the authors. I believe the manuscript could be accepted for publication after addressing the following comment.

The authors claim that the linewidth of the phonon modes is modified on structuring the MoO₃ crystal. The authors estimated the lifetime of the polariton modes for non-patterned MoO₃ and ribbon arrays and reported a 20%–57% reduction to support their claim. I believe, this estimation is cumulative effect over the modified (edges) and unmodified regions. The authors modified the dielectric permittivity of the medium to fit the polariton modes' linewidths. However, the modification in the dielectric permittivity won't be uniform over the whole width of the grating. The accurate estimation of the phonon linewidth can only be seen by the spatially resolved Raman scattering measurements. I suggested the authors to report the phonon linewidth modification from the spatially resolved Raman measurements. This information could be useful for fundamental studies and the technological development of the 2D material for photonic and optoelectronic devices development.

R: In our analysis, the polariton lifetimes were extracted either from the experimental reflectance spectra or near-field images, without any modification made to the dielectric permittivity of the α -MoO₃. Specifically, we extracted the lifetime of the HPhPs in the α -MoO₃ nanoribbon arrays by fitting the reflectance spectrum using a Fano lineshape. Next, using the resonant frequencies of the α -MoO₃ nanoribbon arrays as the excitation frequencies of the incident light, we performed the near-field nanoimaging measurements in the unpatterned region of the same α -MoO₃ flake and extracted the corresponding lifetimes of HPhPs according to the fringes of the optical near-field image. Finally, by comparing the polariton lifetimes extracted from the far-field reflectance spectra of the α -MoO₃ nanoribbon arrays with those obtained from the near-field images of the unpatterned α -MoO₃, we found that the polariton lifetime was reduced by 20%–57% in the ribbon arrays.

The linewidth of the phonon modes in the α -MoO₃ van der Waals (vdW) crystal is directly associated with the loss of phonon polaritons (PhPs). This is because the PhPs are hybridized modes induced by coupling of the incident photon with the intrinsic phonons. For the PhPs, they are sustained within a broad frequency bands bracketed by the transverse and longitude optical phonon frequencies. Therefore, phonon modes with a long lifetime, *i.e.*, a small linewidth, can

give rise to PhPs with small linewidth.

As pointed out by the reviewer, the estimation of PhPs linewidth can indeed be influenced by the cumulative effect of modified edges and unmodified regions of the α -MoO₃ flake. In particular, damage to α -MoO₃ is more likely to occur at the ribbon edges, which could potentially increase the phonon linewidth and result in a broadening of the PhPs linewidths.

As suggested by the reviewer, we evaluated the phonon linewidth, γ , of α -MoO₃ by performing spatially-resolved two-dimensional (2D) Raman mapping on both patterned and unpatterned α -MoO₃ regions using a confocal Raman spectrometer (Renishaw inVia Reflex). A typical Raman spectrum of the unpatterned α -MoO₃ flake is shown in Figure R1a. The linewidths of the phonon modes at 820 cm⁻¹ and 995 cm⁻¹, which originate from the vibrational modes along the [100] and [010] crystallographic directions, respectively, can be readily obtained by fitting the spectrum with a Lorentzian lineshape. In the Raman mapping measurement, the Raman spectra at each position were obtained by scanning the sample under the laser spot. The 2D pseudo-colored images shown in Figure R1b and R1c visualize the spatial distribution of the phonon linewidths over the unpatterned region. The phonon linewidth is rather homogeneous, with $8.0 \text{ cm}^{-1} \leq \gamma \leq 9.0 \text{ cm}^{-1}$ for the mode at 820 cm⁻¹ and $3.0 \text{ cm}^{-1} \leq \gamma \leq 3.5 \text{ cm}^{-1}$ for that at 995 cm⁻¹, respectively.

Figure R1. Raman spectroscopic mapping on unpatterned α -MoO₃ flake. (a) Typical Raman spectrum of the unpatterned α -MoO₃ flake. Inset: enlarged spectrum from 980 cm⁻¹ to 1010 cm⁻¹. The symbols correspond to experimental data while the red lines are Lorentzian lineshape fittings on the experimental spectra. (b, c) Two-dimensional Raman mapping on the unpatterned α -MoO₃ flake. The mappings correspond to linewidths of the phonon modes at 820 cm⁻¹ (b) and 995 cm⁻¹ (c), respectively. The polarization of the incident light is parallel to the [100] crystallographic direction. The mapping was conducted over an area of 50 $\mu\text{m} \times 50 \mu\text{m}$ (red and blue regions in (b) and (c), respectively). The scale bars are 20 μm .

Afterwards, we conducted 2D Raman mapping on α -MoO₃ nanoribbons with widths of $w = 1.0, 2.0, 2.5,$ and $3.0 \mu\text{m}$. The mapping was performed to enclose the ribbon edges, as shown in Figure R2 (pseudo-colored regions). Compared to the unpatterned flake, the overall linewidths of

phonon modes at 820 cm^{-1} and 995 cm^{-1} in these ribbons increased by 4.6% to 14.1% (Table R1). Notably, the ribbon edges exhibited larger linewidths than the center of the ribbon (Figure R2a–R2d) due to defects and impurities introduced during fabrication processes. Therefore, as pointed out by the reviewer, the linewidths of phonon modes are not uniform across the nanoribbon. Additionally, the phonon linewidths increased significantly in the nanoribbon region, particularly at the ribbon edges, compared to the unpatterned region. This broadening contributes to the reduced lifetime of the HPhPs in the nanoribbons.

Figure R2. Raman spectroscopic mapping of α - MoO_3 nanoribbons with widths of $1.0\ \mu\text{m}$ (a), $2.0\ \mu\text{m}$ (b), $2.5\ \mu\text{m}$ (c), and $3.0\ \mu\text{m}$ (d). The mappings correspond to linewidths of the phonon modes at $820\ \text{cm}^{-1}$ and $995\ \text{cm}^{-1}$, respectively. The polarization of the incident light is parallel to the $[100]$ crystallographic direction. The mapping was conducted to enclose a typical ribbon (pseudo-colored regions shown in (a) to (d)). The scale bars are $3\ \mu\text{m}$.

Table R1. Comparison of the linewidths of phonon modes measured from the unpatterned α -MoO₃ flake and 1D ribbon arrays with long axes parallel to [001] crystallographic direction.

Raman Shift (cm ⁻¹)	A: linewidth from unpatterned α -MoO ₃ (cm ⁻¹)	*B: linewidth from 1D ribbon arrays (cm ⁻¹)				Difference: (B-A)/A			
		Width of Ribbon (μ m)				Width of Ribbon (μ m)			
		1.0	2.0	2.5	3.0	1.0	2.0	2.5	3.0
820	8.5	9.2	9.2	9.3	9.7	8.2%	8.2%	9.4%	14.1%
995	3.25	3.6	3.4	3.7	3.5	10.7%	4.6%	13.8%	7.6%

*Maximum value from the Raman mapping on a specific nanoribbon shown in Figure R2 is used for the linewidths of the phonon modes.

It should be noted that due to the diffraction limit, the spatial resolution of the 2D Raman mapping is still limited to larger than 1 μ m. In our future study, we will employ tip-enhanced Raman spectroscopy to reveal more details on the broadening of the phonon modes at the ribbon edges, which we believe can help further the understanding on the mechanisms of the HPhPs lifetimes in the nanoribbon arrays.

As suggested by the reviewer, a discussion on the phonon linewidth modification from the spatially resolved Raman measurements is added in paragraph 3 in page 8 as,

"...The reduction in lifetime of polariton modes in α -MoO₃ ribbons is attributed to defects or impurities at the rough edges of the ribbons introduced during the fabrication process. These imperfections can create additional scattering centers for HPhPs. Furthermore, they will also lead to broadening of the pristine phonon modes. Raman spectroscopic characterizations show that, compared to the unpatterned region, the overall linewidths of the phonon modes in the ribbons increased by 4.6%–14.1% (Supplementary Note 6, Figure 10, Figure 11, and Table 2)...."

Figure R1 and R2 has been added as Supplementary Figure 10 and 11 in the Supplementary Information part. Table R1 is added as Supplementary Table 2 in the Supplementary Information part. In addition, a discussion on Raman mapping of the unpatterned α -MoO₃ and 1D ribbon arrays is added as Supplementary Note 6 in the Supplementary Information part.